# The *Mycobacterium ulcerans* toxin mycolactone causes destructive Sec61-dependent loss of the endothelial glycocalyx and vessel basement membrane to drive skin necrosis

**Louise Tzung-Harn Hsieh[1†], Belinda S Hall[1†], Jane Newcombe[1], Tom A Mendum[1], Sonia Santana Varela[1], Yagnesh Umrania[2], Michael J Deery[2], Wei Q Shi[3], Josué Diaz-Delgado[4], Francisco J Salguero[5], Rachel E Simmonds[1]***

[1]Discipline of Microbes, Infection & Immunity, School of Biosciences, Faculty of Health and Medical Sciences, University of Surrey, Guildford, United Kingdom; [2]Cambridge Centre for Proteomics, University of Cambridge, Cambridge, United Kingdom; [3]Department of Chemistry, Ball State University, Muncie, United States; [4]Texas A&M Veterinary Medical Diagnostic Laboratory, College Station, United States; [5]United Kingdom Health Security Agency, UKHSA-Porton Down, Salisbury, United Kingdom

**\*For correspondence:**
rachel.simmonds@surrey.ac.uk

[†]These authors contributed equally to this work

**Competing interest:** The authors declare that no competing interests exist.

## eLife Assessment

The toxin mycolactone is produced by Mycobacterium ulcerans which is responsible for the Buruli ulcer lesions. The authors performed a **valuable** study showing the effects of mycolactone on blood vessel integrity. This **convincing** data provides new therapeutic targets to accelerate the healing of Buruli ulcer lesions.

**Abstract** The drivers of tissue necrosis in *Mycobacterium ulcerans* infection (Buruli ulcer disease) have historically been ascribed solely to the directly cytotoxic action of the diffusible exotoxin, mycolactone. However, its role in the clinically evident vascular component of disease aetiology remains poorly explained. We have now dissected mycolactone's effects on human primary vascular endothelial cells in vitro. We show that mycolactone-induced changes in endothelial morphology, adhesion, migration, and permeability are dependent on its action at the Sec61 translocon. Unbiased quantitative proteomics identified a profound effect on proteoglycans, driven by rapid loss of type II transmembrane proteins of the Golgi, including enzymes required for glycosaminoglycan (GAG) synthesis, combined with a reduction in the core proteins themselves. Loss of the glycocalyx is likely to be of particular mechanistic importance, since knockdown of galactosyltransferase II (beta-1,3-galactotransferase 6; B3GALT6), the GAG linker-building enzyme, phenocopied the permeability and phenotypic changes induced by mycolactone. Additionally, mycolactone depleted many secreted basement membrane components and microvascular basement membranes were disrupted in vivo during *M. ulcerans* infection in the mouse model. Remarkably, exogenous addition of laminin-511 reduced endothelial cell rounding, restored cell attachment and reversed the defective migration caused by mycolactone. Hence supplementing mycolactone-depleted extracellular matrix may be a future therapeutic avenue, to improve wound healing rates.

## Introduction

Buruli ulcer (BU) is a neglected tropical disease caused by subcutaneous infection with *Mycobacterium ulcerans,* characterised by development of large, painless plaques or open lesions, often associated with oedema. The disease is most common in West Africa but is also found in other tropical and subtropical regions including Australia. Although the lesion can be sterilised by a minimum 2-month antimicrobial treatment course with rifampicin and clarithromycin, the wounds can take up to a year to heal and can lead to permanent disfigurement, especially when diagnosed late (*Yotsu et al., 2018*). The polyketide-derived toxin mycolactone, generated by *M. ulcerans*, is the critical driver of BU pathogenesis (*George et al., 1999*; *Sarfo et al., 2016*). Continuous production of this virulence factor causes widespread coagulative necrosis and fibrin deposition in patient skin tissue, as it diffuses through tissue away from the infecting bacteria. Mycolactone is also responsible for the restricted immune response seen in BU. As well as showing long-term cytotoxicity to immune cells (*George et al., 1999*), mycolactone causes a rapid suppression of antigen presentation, co-stimulation and cytokine secretion at low doses (*Guenin-Macé et al., 2011*; *Simmonds et al., 2009*).

Many clinical features of BU can be attributed to the inhibitory action of mycolactone on the Sec61 translocon (*Baron et al., 2016*; *Hall et al., 2014*), the complex that translocates most membrane, secretory and organellar polypeptides into the endoplasmic reticulum (ER; *O'Keefe et al., 2022*). In co-translational translocation, nascent proteins are targeted to the ER surface by a signal peptide sequence at the N-terminus; the interaction between the signal peptide and the pore-forming protein of the translocon, Sec61α, opens a central channel that allows access to the ER lumen and a lateral gate through which transmembrane sequences can enter the membrane (*Voorhees and Hegde, 2016*). Mycolactone docks to Sec61, preventing signal peptide engagement and locking the translocon in an inactive state with the lateral gate open but the channel blocked (*Gérard et al., 2020*). The biogenesis of most secretory proteins and Type I and II membrane proteins is inhibited by mycolactone, while polytopic membrane proteins are largely unaffected (*McKenna et al., 2017*; *Morel et al., 2018*). Type III and tail-anchored proteins, which utilise alternative translocation pathways (*O'Keefe et al., 2021*), are also resistant to mycolactone (*Hall et al., 2014*; *McKenna et al., 2017*; *Grotzke et al., 2017*). The proteins whose translocation into the ER is blocked are synthesized in the cytosol where they are degraded by the proteosome (*Hall et al., 2014*) and selective autophagy (*Hall et al., 2022*; *Gama et al., 2014*). Sec61 blockade induces an integrated stress response by activation of eIF2α kinases (*Morel et al., 2018*; *Ogbechi et al., 2018*) and an increase in autophagic flux (*Hall et al., 2022*) and without resolution the cells eventually undergo apoptosis (*Ogbechi et al., 2018*; *Bieri et al., 2017*). The time from initial exposure to cell death varies between cell types, but for most human cells takes 3–5 days.

We have previously shown that endothelial cells are particularly sensitive to mycolactone. At low nanomolar concentrations, mycolactone depletes the anticoagulant receptor thrombomodulin (*Ogbechi et al., 2015*) and junction proteins (*Hsieh et al., 2022*). It also increases the permeability of monolayers formed from endothelial cells derived from both vascular and lymphatic origin (*Hsieh et al., 2022*). While thrombomodulin depletion has also been observed in BU patient skin biopsies (*Ogbechi et al., 2015*), this seems not to be the cause of the widespread fibrin deposition commonly seen within the skin tissue. Instead, this is linked to aberrant staining for the extrinsic clotting pathway initiator tissue factor (*Hsieh et al., 2022*). Tissue factor is normally located in the sub-endothelium where it is segregated from both the plasma proteins that drive coagulation and the surrounding dermal tissue (*Butenas et al., 2009*). However, in BU patients, tissue factor was observed within the connective tissue distant from vessels and this spatially associated with fibrin deposition and early signs of necrosis (*Hsieh et al., 2022*). Our working model leading up to the current work was that mycolactone action at Sec61 in endothelial cells leads to vascular dysfunction and promotes the pathogenesis of BU. The current work seeks to explore the molecular mechanisms driving these events.

The integrity of the endothelium greatly depends on adequate production and maintenance of the extracellular matrix (ECM) (*Davis and Senger, 2005*), junctional complexes (*Wallez and Huber, 2008*) and the glycocalyx, a highly charged coating of proteoglycans, glycolipids, glycoproteins and glycosaminoglycans (GAG) including heparan sulphate (HS), chondroitin sulphate (CS) and hyaluronic acid (*Zeng et al., 2012*) covering the luminal side of the endothelium (*Reitsma et al., 2007*). The enzymatic glycosylation of heparan and chondroitin sulphate is initiated in the Golgi apparatus by

transferase enzymes. This is also the site of the isomerisation and sulfation reactions needed to achieve the rich diversity of GAGs expressed at the cell surface (*Yilmaz et al., 2019*). The glycocalyx acts as an exclusion zone for blood cells and controls interactions with platelets, blood clotting factors and immune cells as well as modulating fluid exchange and acting as a sensory system for the endothelial monolayer (*Yilmaz et al., 2019*). On the basal side of the endothelium, is the basement membrane (BM), an ECM consisting of collagen type IV and laminins, crosslinked by perlecan, a HS proteoglycan, and/or nidogens (*Yousif et al., 2013*). This sheet-like network forms a scaffold that interacts with integrins on the cell surface, controlling structural stability, cell adhesion and angiogenesis as well as preventing leukocyte extravasation (*Yousif et al., 2013*; *Song et al., 2017*). Production of these complex structures, which preserve and regulate the barrier between blood and tissue, relies heavily on Sec61-dependent proteins.

In order to determine the molecular mechanisms driving mycolactone-induced endothelial cell dysfunction, we have undertaken a detailed phenotypic and proteomic study of the changes it induces both in vitro and in vivo. Using primary human dermal microvascular endothelial cells (HDMEC), we found that, as well as increasing monolayer permeability, mycolactone caused rapid changes in endothelial cell morphology and migration, accompanied by loss of glycocalyx, adhesion and ECM proteins. Notably, structurally unrelated Sec61 inhibitors, Ipomoeassin F, and its derivatives induced comparable phenotypes in a similar time frame, highlighting the Sec61 dependency of ECM composition and function. We have dissected the roles of these different components in the response to mycolactone and found that loss of an enzyme critical for GAG biosynthesis phenocopied the changes seen in cell morphology and monolayer permeability. On the other hand, the effects on cell adhesion and migration were dependent on ECM interactions and could be ameliorated by application of exogenous laminin-511. Hence the current work presents a novel pathogenic mechanism in BU, driven by Sec61-dependent effects on endothelial cells.

## Results
### Sec61 blockade impacts endothelial cell morphology and adhesion

We recently observed that mycolactone induces morphological changes in primary endothelial cells in vitro, leading to a dose-dependent increase in monolayer permeability at 24 hr (*Hsieh et al., 2022*). To understand the longer-term effects of mycolactone, we performed time-lapse imaging of HDMECs exposed to mycolactone (*Figure 1—video 1*) or solvent (DMSO) control (*Figure 1—video 2*) every 30 min for 48 hr. As in previous observations, the cells began to take on an 'elongated' phenotype after 8 hr. The proportion of elongated cells increased with time (*Figure 1A*) and after 24 hr exposure, approximately half the cells (51.63 ± 2.89%) had this phenotype. The average ratio of cell length to width doubled in 16 hr, and quadrupled after 24 hr exposure (*Figure 1C*). At 24 hr, a small proportion (9.73 ± 4.01%) had acquired a rounded appearance (*Figure 1B*) similar to that reported for mycolactone exposure of fibroblasts (*Gama et al., 2014*) and epithelial cells (*Guenin-Macé et al., 2013*). Notably, these cells retained the ability to reattach to the culture vessel (*Figure 1—video 1*), in line with their continued viability in this time window (*Ogbechi et al., 2015*). However, after this time their ability to re-adhere declined and the proportion of detached cells steadily increased. Although the number of rounded cells increased between 24 and 48 hr, the elongated phenotype remained predominant at this time point (*Figure 1B*).

In order to confirm that these phenotypes were relevant to biologically derived mycolactone, we compared the response to the synthetically made mycolactone to that of mycolactone A/B extracted from *M. ulcerans* bacteria. The preparations showed equivalent potency against HDMEC (*Figure 1—figure supplement 1*) and caused similar changes in phenotype in live cell imaging assays (*Figure 1—figure supplement 2* and *Figure 1—figure supplement 3*).

Next, we investigated how mycolactone affected HDMEC migration using scratch assays. While control cells were successfully able to close the scratch area within 24 hr, mycolactone-exposed cells displayed a gradual cessation in migration into the cell-free gap (*Figure 1D*). Thus, while at 16 hr similar numbers of cells had migrated into the scratch regardless of treatment, no further migration could be detected at 24 hr in the presence of mycolactone (*Figure 1D*). However, it should be noted that mycolactone has previously been reported to cause cell cycle arrest (*George et al., 1999*), which can be a confounding factor in such migration assays and may explain this finding.

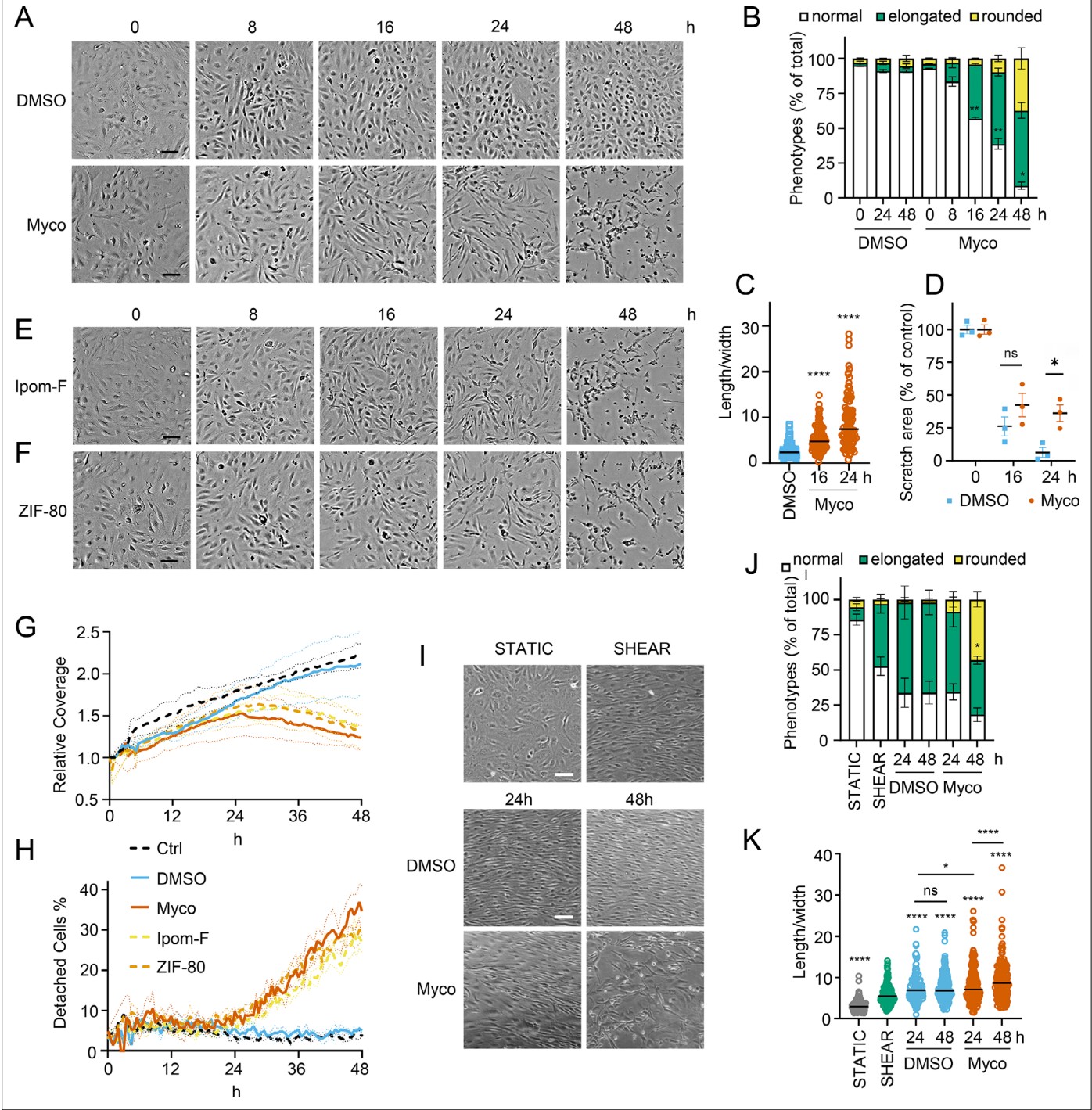

**Figure 1.** Sec61 inhibition alters primary human endothelial cell morphology and adhesion under static and shear conditions. HDMECs were exposed to 10 ng/mL mycolactone (Myco), 0.02% DMSO, 400 nM Ipomoeassin F or 20 nM ZIF-80. (**A–C**). Mycolactone-treated cells were imaged at indicated times in (**A**) and cell numbers of each phenotype (i.e. normal, elongated or rounded) were counted and presented as a percentage of total cell number per field in (**B**) (mean ± SEM of three independent experiments, **, p<0.01, *, p<0.05. (**C**) Length and width of each cell exposed to mycolactone for 16 and 24 hr or DMSO for 24 hr per image were measured and presented as a ratio. Data is representative of three independent experiments. ****, p<0.0001 (**D**) A scratch was introduced to a HDMEC monolayer prior to the treatment and visualised at 0, 16, 24 hr. The scratch area is presented as a percentage of the value obtained at 0 hr (mean ± SEM of three independent experiments) ns, not significant, *, p<0.05. Cells exposed to an alternative Sec61 inhibitor ipomoeassin F (IpomF) (**E**) or ZIF-80 (**F**) were imaged at indicated times. Images are representative of three independent experiments. Scale bar = 100 μm. (**G–H**). Live cell imaging was performed with the zenCELL Owl incubator microscope every 30 min over 48 hr. Algorithms of cell coverage (**G**), detached cell numbers (**H**) per time point from three independent experiments are summarised as mean ± SEM. Data presented as

*Figure 1 continued on next page*

*Figure 1 continued*

cell coverage relative to the value obtained from initial time point (**G**) or a % of detached cells to total cell number (**H**) of each condition. (**I**) Confluent HDMECs under uniaxial shear stress (SHEAR) or not (STATIC) for 24 hr were then exposed to 0.02% DMSO or 10 ng/ml mycolactone. Phase-contrast images were taken 0, 24 and 48 hr later following return to the same conditions. Data is representative of three independent experiments. (**J**). Cell numbers of each phenotype at different time points, presented as a percentage of total cell number per field in (**I**), showing mean of three independent experiments ± SEM, * p<0.05. (**K**). Length and width measurements of cells exposed to mycolactone or DMSO under shear stress conditions for 24 and 48 hr presented as a ratio. Data is representative of three independent experiments. ****, p<0.0001. Statistical analysis was performed by two-way ANOVA (panel **B**, **D and J**) or one-way ANOVA with Dunnett's (panel **C**) or Tukey's (panel **D**) correction in GraphPad Prism Version 9.4.1 and 10.2.3. Panel **K**) was analysed using a mixed-effects model with Tukey's correction for multiple comparisons.

The online version of this article includes the following video, source data, and figure supplement(s) for figure 1:

**Source data 1.** Data points used to generate the graphs in *Figure 1B, C, D, J and K*.

**Figure supplement 1.** Dose response of HDMEC to mycolactone produced synthetically (Myco-Syn) or purified from natural sources (Myco-Bio).

**Figure supplement 2.** Images of HDMEC treated with 10 ng/ml synthetic (Syn) or natural (Nat) or 0.02% DMSO.

**Figure supplement 3.** Quantification of cell phenotypes in *Figure 1—figure supplement 2*.

**Figure supplement 4.** Quantification of coverage and detachment in *Figure 1—figure supplement 2* using zenCELL Owl algorithms of cell coverage (Left panel), detached cell numbers (Right panel).

**Figure 1—video 1.** Timelapse video of HDMECs exposed to 10 ng/mL mycolactone.
https://elifesciences.org/articles/86931/figures#fig1video1

**Figure 1—video 2.** Timelapse video of HDMECs exposed to 0.02% DMSO.
https://elifesciences.org/articles/86931/figures#fig1video2

To determine whether Sec61 inhibition by mycolactone was driving these abnormal phenotypes, we exposed endothelial cells to Ipomoeassin F or its more potent derivative, ZIF-80 (*Zong et al., 2020*). These are structurally distinct to mycolactone but inhibit Sec61α in a very similar manner since they compete for the same binding site (*Zong et al., 2019*; *Pauwels et al., 2021*). Importantly, both compounds phenocopied the 'elongated' appearance preceding detachment in HDMEC within 24 hr (*Figure 1E & F*). Unbiased analysis of time-lapse data using zenCELL owl built-in algorithms allowed continuous estimation of cell coverage and detachment, although it could not be trained to recognise the elongated phenotype. As expected, cell coverage increased with time under control conditions, while the proportion of detached cells remained constant at approximately 5%. However, all three Sec61 inhibitors showed similar effects on both readouts (*Figure 1G & H*), with a similar effect of biologically purified mycolactone (*Figure 1—figure supplement 4*). Interestingly, both measures remained similar to the control for approximately 24 hr, after which cell coverage declined with a corresponding increase in cell detachment. Taken together, this data strongly supports that these changes are driven by Sec61 inhibition and that endothelial cell homeostasis is dependent on adequate Sec61 function.

As the responses of endothelial cells grown in static culture plates may not accurately reflect in vivo behaviour where the cells lining blood vessels are subject to shear stress, HDMEC were cultured on an orbital shaker to more closely mimic conditions experienced under flow. Cells were imaged at the periphery of wells where they experience uniaxial shear stress. Cells were grown to confluency and incubated on a rotary shaker for 24 hr then mycolactone was added. Cell elongation and a uniform alignment of the endothelial cells was maintained for 48 hr in the presence of DMSO, as expected (*Figure 1I*). By contrast, cells exposed to mycolactone became increasingly disorganised over time and by 48 hr a significant proportion were rounded and detached (42.6%, p<0.05) (*Figure 1I*). Although mycolactone did not cause a change in the proportion of elongated cells under these conditions, the pattern of cell rounding was similar to that seen in static cultures (*Figure 1J*). There was no detectable difference in the length:width ratio at 24 hr, but by 48 hr mycolactone-exposed HDMEC that remained adhered were significantly elongated compared the DMSO control (p<0.001; *Figure 1K*). Thus, although the kinetics are slightly different, the phenotypic changes induced by mycolactone in static culture are reproducible in endothelial cells under flow conditions.

To establish the in vivo relevance of these findings, we performed fibrinogen immunostaining in the pre-ulcerative mouse footpad model of *M. ulcerans* infection (*Figure 2*, *Figure 2—figure supplement 1*). Fibrinogen is a high molecular weight (~330 kDa) plasma protein that is normally retained within the lumen of intact vessels and, indeed, in uninfected (vehicle control) mouse feet, fibrinogen

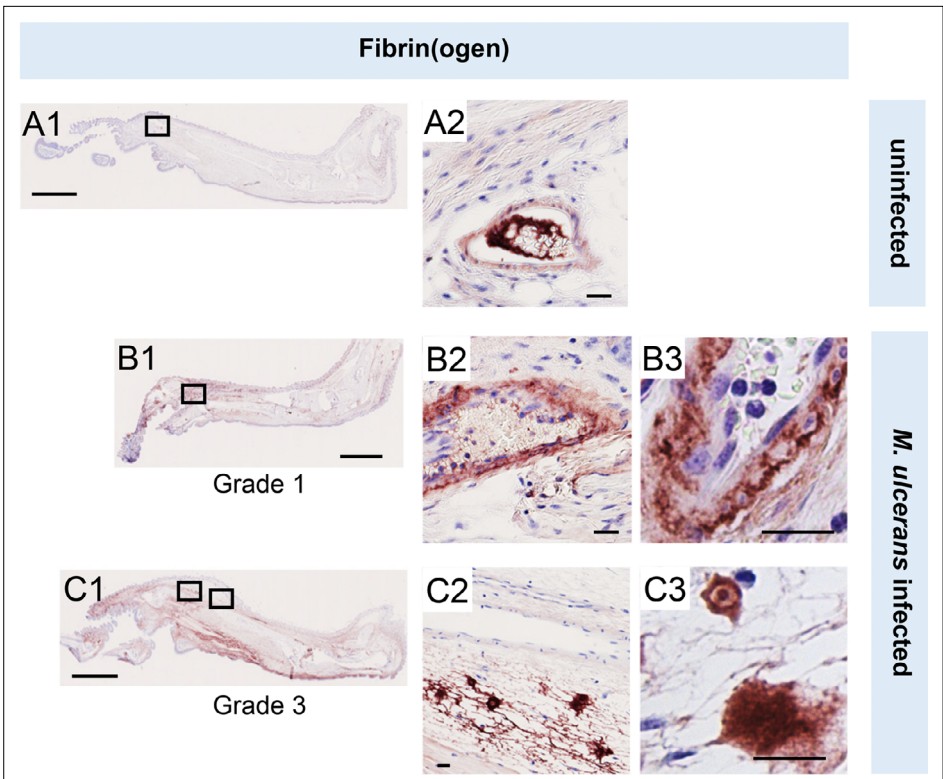

**Figure 2.** Fibrinogen penetrates the vascular wall at early stages of mouse infection. Immunohistochemistry for fibrin(ogen) in the feet of C57BL/6 J mice that received vehicle control (PBS) (A1-2) or intradermal injection of $1 \times 10^5$ colony forming units *M. ulcerans* at 21 (Grade 1; B1-3) or 28 days (Grade 3; C1-3) post-infection. Positive fibrin(ogen) staining is brown in colour, the haematoxylin counterstain is purple. Scale bars in A1, B1, and C1: 2 mm; all others: 20 μm. Data representative of three mice at each infection grade, in two independent infections.

The online version of this article includes the following figure supplement(s) for figure 2:

**Figure supplement 1.** Sections stained for acid fast bacilli with Ziehl Neelson stain from mice infected for 21 days (**G1**) or 28 days (**G2**).

**Figure supplement 2.** *M.ulcerans* infected mouse feet stained with isotype control antibody.

was rarely detected, and then only within the vessel lumen (*Figure 2A*). In contrast, at 21 days post infection (Grade 1 lesions; metatarsal thickness increase ~10%), fibrin(ogen) was seen within the blood vessel wall surrounding the endothelium (*Figure 2B*). After 28 days (Grade 2/3 lesions, metatarsal thickness increase 50–100%), widespread fibrin(ogen) staining was seen outside blood vessels within the dermis, in foci consistent with its conversion to insoluble fibrin (*Figure 2C*). The lack of signal in isotype control-stained tissue (*Figure 1—figure supplement 2*) confirms the specificity of staining. This penetration of fibrinogen between the endothelial monolayer lining the vessel, then through the vessel wall and conversion to fibrin by other components of the coagulation cascade within deeper tissue is consistent with our previous findings in human BU patient punch biopsies (*Hsieh et al., 2022*). Furthermore, the changes in endothelial cell morphology and monolayer integrity described here and previously *Hsieh et al., 2022* demonstrates that the extravascular deposition of fibrin is an early feature of infection.

## Mycolactone predominantly targets proteins involved in glycosylation and adhesion

While proteomic studies of mycolactone action have been performed previously (*Baron et al., 2016*; *Morel et al., 2018*; *Grotzke et al., 2017*; *Gama et al., 2014*), these have used whole cell lysates, leading to systematic limitations in detection of membrane and secreted proteins, due to their relatively low abundance compared to cytosolic proteins. Therefore, to understand the molecular

mechanisms driving the pathogenic phenotypic changes in endothelial cells, we instead used a total membrane proteomics approach to enrich for the Sec61 substrates that are targeted by mycolactone.

We isolated total membrane fractions from HDMECs exposed to DMSO or mycolactone for 24 hr and analysed them by tandem mass tagging (TMT) multiplex LC/MS over biological triplicates (*Figure 3A*). A total of 6649 proteins were detected, of which 482 were significantly downregulated and 220 upregulated by mycolactone (>2 Fold change, p<0.05; *Figure 3B*, *Figure 3—source data 1*). Among the total proteins discovered, 36.9% were trafficked via the secretory/endolysosomal pathways that primarily depend on the Sec61 translocon (*Figure 3C*). This group represented 84.6% of the downregulated but only 23.7% of the upregulated proteins. As predicted, membrane proteins were the most affected in the downregulated group, with little effect on cytoplasmic, cytoskeletal, mitochondrial or nuclear proteins. The downregulated fraction included previously published endothelial targets of mycolactone including coagulation regulators thrombomodulin (TM), von Willebrand Factor (vWF), platelet endothelial cell adhesion molecule (CD31), endothelial protein C receptor and tissue factor pathway inhibitor (TFPI) and cell junction components tyrosine protein kinase receptor TIE1, angiopoietin-1 receptor (TEK), cadherin 5 (CDH5), junctional adhesion molecule 3 (JAM-3) and catenin β1 (*Ogbechi et al., 2015*; *Hsieh et al., 2022*; *Figure 3—figure supplement 1*), validating our dataset.

As seen in previous proteomic studies and in vitro translocation assays (*McKenna et al., 2017*; *Morel et al., 2018*; *Demangel and High, 2018*; *McKenna et al., 2016*), mycolactone preferentially targeted secreted and single pass type I and type II membrane proteins in endothelial cells, with no effect on the EMC-dependent Type III proteins or the GET pathway-dependent tail-anchored proteins (*Figure 3D*). A small number (51 out of 606 detected) of multi-pass membrane proteins were also >twofold downregulated by mycolactone (*Figure 3—source data 1*). This group was relatively enriched for signal peptide-bearing proteins (42% vs 4% amongst unchanged and upregulated multi-pass proteins; *Figure 3E*). The rules governing sensitivity of this subgroup to mycolactone appear to be similar to those reported for single pass type I proteins (*Morel et al., 2018*), with higher signal peptide hydrophobicity and a shorter distance between the signal peptide and first transmembrane domain being associated with increased resistance to the effects of mycolactone (*Figure 3—figure supplement 2*). Of the remaining mycolactone-sensitive multi-pass proteins, 80% contained at least one long loop (>50 aa) between transmembrane domains. Among the upregulated proteins, 88% of the integral membrane proteins were multi-pass membrane proteins, and only one of the predicted single pass proteins contained a signal peptide. Likewise, the four upregulated secreted proteins identified are all secreted by non-conventional pathways.

Overall, the data support the recently described model for the biogenesis of multi-pass proteins whereby the majority of multimembrane spanning proteins utilise an alternative translocon that includes Sec61 and the PAT, GEL and BOS complexes but, crucially, bypasses the lateral gate, instead relying on generation of a lipid-filled cavity on the opposite side of Sec61 (*Smalinskaitė et al., 2022*; *Sundaram et al., 2022*). Here, only those multi-pass proteins possessing a signal peptide or long internal loops require insertion into the membrane via the Sec61 channel, and therefore only these are sensitive to mycolactone.

Our membrane targeted approach identified a higher number of Sec61-dependent proteins in our control cells compared to previous studies (*Baron et al., 2016*; *Morel et al., 2018*; *Grotzke et al., 2017*) thus achieving our goal of wider capture of mycolactone-sensitive proteins. Moreover, when compared to siRNA-based Sec61α knockdown in Hela cells, despite the differences in cell type and methodology, 100 of the downregulated proteins were common to both datasets (*Figure 3F*; *Nguyen et al., 2018*). While possession of a signal peptide or anchor appears to be crucial to mycolactone sensitivity (*Morel et al., 2018*), overall, no specific signal peptide sequence features were associated with downregulation. In keeping with this, there was very little overlap between mycolactone downregulated proteins and those lost following knockdown of translocon-associated proteins TRAPβ or knockout of Sec62/Sec63 (*Figure 3—figure supplement 3*; *Nguyen et al., 2018*; *Schorr et al., 2020*), which assist gating of the translocon by weak signal peptides. Thus, as suggested by analysis of the structure of the inhibited translocon (*Gérard et al., 2020*), mycolactone acts via direct interaction with the Sec61α signal peptide binding site rather than through interference with accessory proteins.

Gene ontology (GO) analysis of mycolactone-upregulated proteins supported previous observations by ourselves and others of cellular stress responses, with significant enrichment of terms

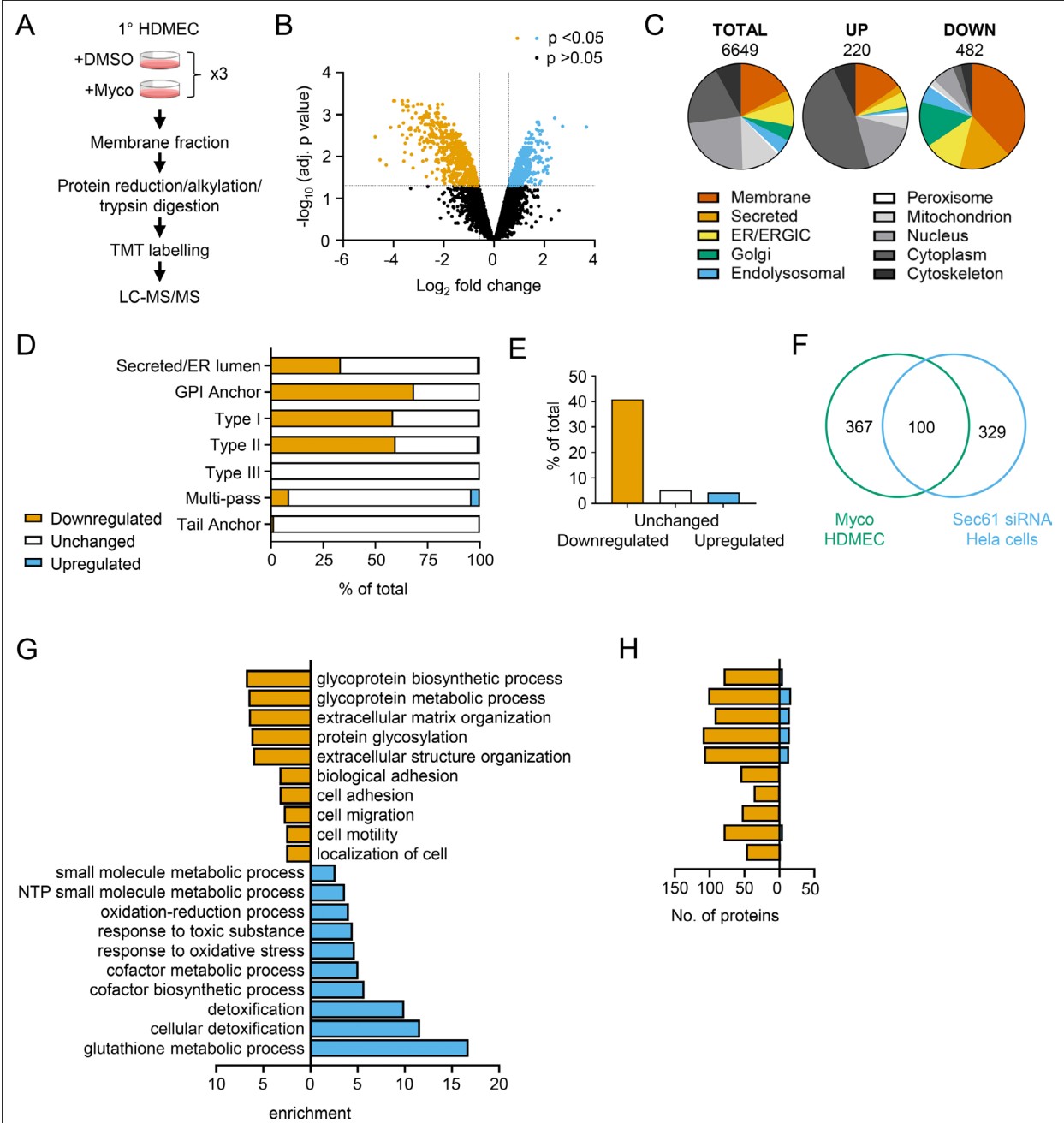

**Figure 3.** Mycolactone causes loss of proteins associated with glycosylation, adhesion and migration in primary human endothelial cells. (**A**) Workflow for isolation and proteomic analysis of HDMEC membrane proteins. In three independent replicates, HDMEC were exposed to 10 ng/ml mycolactone or DMSO for 24 hr, lysed by hypotonic lysis and membrane fractions enriched by differential centrifugation as described in Methods. Acetone precipitated proteins were reduced, alkylated and trypsinised then subjected to TMT labelling for quantitative proteomic analysis by LC-MS/MS. (**B**) Volcano Plot of differential expression between DMSO and mycolactone treated samples, plotting mean fold change against false discovery rate adjusted p-values; orange = downregulated, p<0.05; blue = upregulated, p<0.05; black = p > 0.05 (**C**) Pie charts showing subcellular localisation of proteins in total, >twofold upregulated or downregulated (p<0.05) fractions. (**D**) Quantitation of membrane or secreted proteins according to type: blue = upregulated; white = unchanged; orange = downregulated. (**E**) Percentage of downregulated, unchanged and upregulated multi-pass membrane proteins possessing a signal peptide. (**F**) Overlap between mycolactone downregulated endothelial membrane proteome and Sec61-dependent proteome. Venn diagram created using JVenn (***Bardou et al., 2014***), showing overlap in significantly downregulated proteome between the dataset presented here and those obtained in Hela cells treated with siRNA for Sec61α (***Nguyen et al., 2018***). (**G**) Top significantly over-represented (p<0.05) GO groups in downregulated and upregulated data sets, compared to whole genome. Data generated with WebGestalt. (**H**) Quantitation of numbers of up and down regulated proteins in GO groups identified in (**G**).

*Figure 3 continued on next page*

eLife Research article

Microbiology and Infectious Disease

*Figure 3 continued*

The online version of this article includes the following source data and figure supplement(s) for figure 3:

**Source data 1.** Mass spectrometry proteomics data for membrane fractions of HDMEC exposed to DMSO or Mycolactone for 24 hr.

**Figure supplement 1.** Heat map showing fold change detected in this dataset for previously validated endothelial cell mycolactone targets (*Ogbechi et al., 2018*; *Hsieh et al., 2022*) Dual-colour coding is shown.

**Figure supplement 2.** Left panel: Significant association between multipass protein membrane protein signal peptide (SP) ΔG values and level of downregulation by mycolactone (p<0.05).

**Figure supplement 3.** Left panel: Model depicting assisted and unassisted channel opening by signal peptides.

**Figure supplement 4.** Top significantly over-represented (*P*<0.05) Gene Ontology groups in upregulated data set, compared to whole genome.

associated with oxidative stress and detoxification (*Figure 3G*; *Morel et al., 2018*; *Ogbechi et al., 2018*; *Förster et al., 2020*; *Grönberg et al., 2010*). The upregulated proteins also included several proteins involved in the autophagy pathway, including SQSTM1/p62, which is involved in the cellular response to mycolactone (*Hall et al., 2022*; *Figure 3—figure supplement 1*). However, in the significantly downregulated fraction a distinct pattern emerged, with GO terms associated with glycosylation, matrix organisation, adhesion and cell migration showing the greatest over-representation compared to the whole genome. Within these GO groups, the vast majority of proteins detected in our proteome were downregulated by mycolactone (*Figure 3H*). Similar results were obtained when the downregulated proteins were compared to the total detected proteome (*Figure 3—figure supplement 4*), showing this pattern was not an artefact resulting from membrane enrichment.

## Mycolactone disproportionately targets Golgi-resident proteins involved in glycosylation and glycosaminoglycan chain synthesis leading to the loss of surface GAGs

The Golgi is the site of higher order protein glycosylation and GAG synthesis and, of the intracellular organelles, is the most affected by mycolactone (*Figure 3C*, *Figure 4—figure supplement 1*). The Golgi has a particularly high proportion of type II membrane proteins as the membrane anchor and sequences around it can act as a signal for Golgi retention (*Kikegawa et al., 2018*) and nearly all of these Golgi-expressed type II membrane proteins were significantly downregulated by mycolactone (*Figure 4A*). Interestingly, type II Golgi proteins showed a higher degree of down-regulation by mycolactone than ER or plasma membrane localised type II proteins (*Figure 4B*). This suggests the signals that lead to Golgi localisation may make proteins more sensitive to Sec61 inhibition, although it is equally possible that Golgi proteins are turned over at a higher rate than those at other sites as depletion is generally at the turnover rate (*Ogbechi et al., 2015*). The effect is not due to differences in transmembrane domain hydrophobicity, which shows little variation and has no impact on Type II protein levels in mycolactone-treated cells (*Figure 4—figure supplement 2*).

Detailed analysis of our dataset revealed that targeting of Golgi-localised proteins by mycolactone leads to significantly decreased abundance of multiple enzymes involved in both higher order N- and O-linked glycosylation (*Figure 4C*). However, the biggest impact is seen in GAG production, with the majority of the enzymes involved in GAG synthesis lost in mycolactone treated HDMECs (*Figure 4D*, *Figure 4—figure supplement 3*). All of the 23 proteins in the GAG biosynthetic pathway detected in our analysis are type II membrane proteins and 19 (82%) of these were downregulated by mycolactone (*Figure 4D*), affecting every step of glycosaminoglycan production (*Figure 4—figure supplement 3*). Three of the mycolactone-targeted proteins were involved in initial steps of keratan sulphate formation, four in common synthesis initiation of chondroitin sulphate (CS), dermatan sulphate (DS), and heparan sulphate (HS), six in chain elongation of CS/DS and HS (two and four, respectively), and six in epimerisation or sulfation processes that enhance the structural diversity of CS/DS or HS (*Figure 4D*, *Figure 4—figure supplement 3*).

Given the importance of GAGs to endothelial function and the dramatic loss in GAG biosynthetic enzymes induced by mycolactone, we evaluated surface levels of the predominant endothelial GAGs, HS, and CS, using flow cytometry on HDMECs exposed to mycolactone for 24 hr. As a control, chondroitinase ABC was used to remove surface CS, resulting in fluorescence levels 60% lower than untreated cells. Remarkably, CS fluorescence intensity was even lower in cells exposed to mycolactone (*Figure 4E*). Similarly, using an antibody specific for a neoepitope of HS generated by heparinase

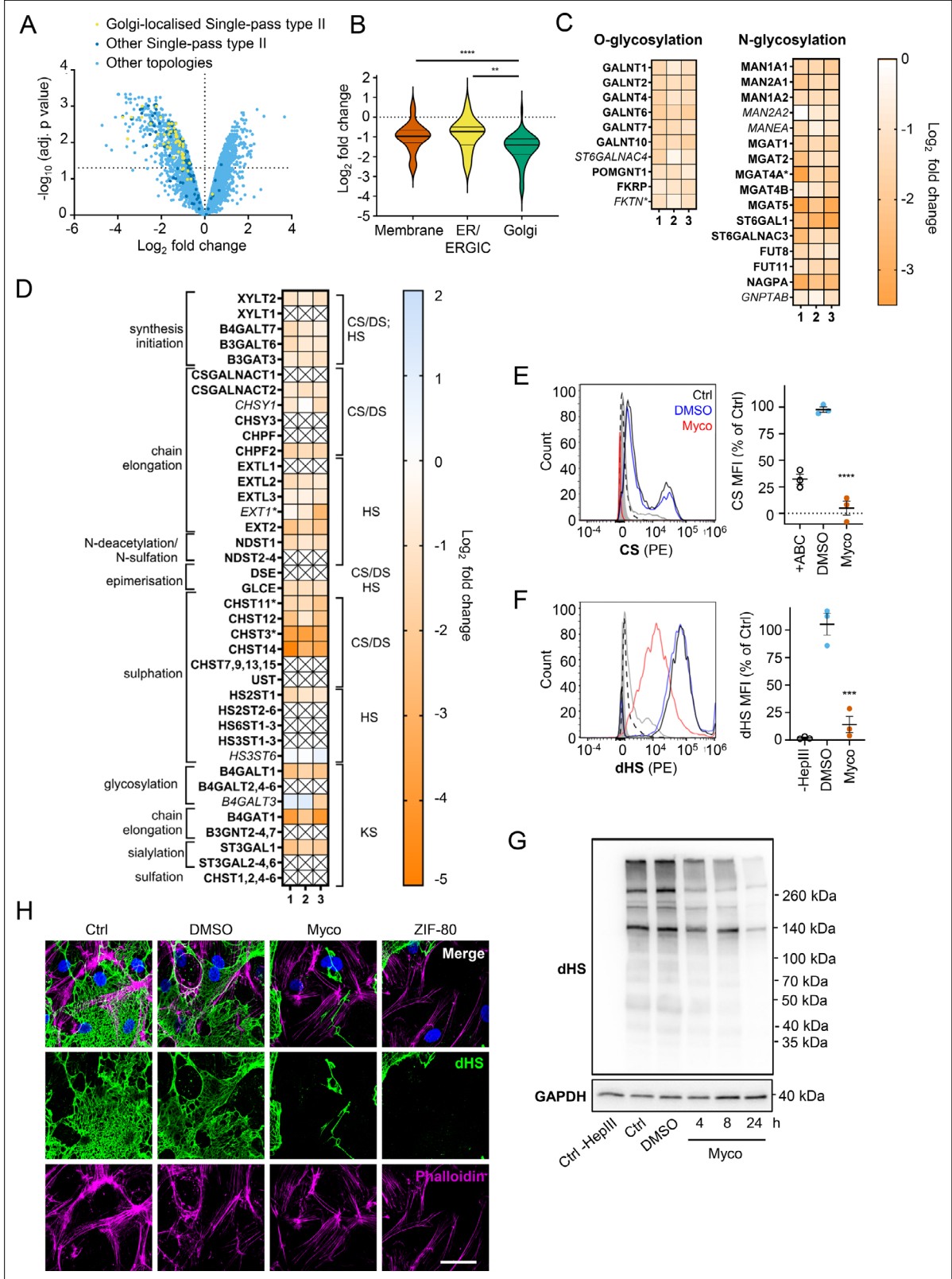

**Figure 4.** Endothelial glycosaminoglycan chain synthesis is blocked by mycolactone in primary human endothelial cells. HDMECs exposed to 10 ng/mL of mycolactone (Myco) or 0.02% DMSO for 24 hr or indicated times were subjected to proteomic analysis (**A–D**), surface immunostaining (**E–F, H**) or immunoblotting (**G**). (**A**) Volcano Plot of differential expression between DMSO and mycolactone-treated samples, plotting mean fold change against false discovery rate adjusted *p*-values. Pale blue = total detected proteins; dark blue = Type II membrane proteins; yellow = Golgi-localised Type II

*Figure 4 continued*

membrane proteins. (**B**) Violin plot showing fold change in protein levels for Type II membrane proteins grouped according to subcellular location. ns, not significant; **, p<0.01; ****, p<0.0001. (**C**) Heat map showing fold change in Golgi-localised O- and N-glycosylation enzymes in mycolactone exposed HDMEC. Dual-colour coding is shown, only one unique peptide detected in asterisks, and significantly downregulated (p<0.05) or not (p≥0.05) in bold or *Italic,* respectively. (**D**) Genes in GAG biosynthesis categorised according to function and side chains of chondroitin sulphate/ dermatan sulphate (CS/DS), heparan sulphate (HS) or keratan sulphate (KS). Heatmap showing Log2 fold change of these genes in response to mycolactone in three independent experiments. Dual-colour coding is shown. Genes undetected are indicated as crossed, only one unique peptide detected in asterisks, and significantly downregulated (p<0.05) or not (p≥0.05) in bold or *Italic,* respectively. (**E–F**) Cells were treated with or without chondroitinase ABC (ABC) or heparinase III (HepIII), immunostained with anti-chondroitin sulphate (CS), anti-Δ-heparan sulphate (dHS) antibodies or the isotype controls for flow cytometry analysis. Histogram plot for single cell population of CS (**E**) and dHS (**F**) and the respective mean fluorescence intensity (MFI) are shown. Unstained untreated cells filled grey; isotype control of untreated cells, dashed line in black; untreated cells incubated with chondroitinase ABC prior to CS staining or without HepIII prior to dHS staining, grey line; untreated cells with CS-PE or dHS-PE, black line; cells exposed to DMSO stained with antibodies, blue line; cells exposed to mycolactone stained with antibodies, red line. MFI is presented as a % of untreated control (mean ± SEM of three independent experiments). **, p<0.01; ***, p<0.001; ****, p<0.0001. (**G**) Cells were lysed, treated with heparinase III and analysed by immunoblotting. HS neoepitopes were visualised with anti- Δ-heparan sulphate (dHS) antibody with the approximate migration of molecular weight markers in kDa. GAPDH used as loading control. Images are representative of three independent experiments. (**H**) Cells were incubated with HepIII, fixed and immunostained with anti-dHS antibody (green), permeabilised and labelled with TRITC-conjugated phalloidin (magenta). Nuclei were stained with DAPI (blue). Images are representative of two independent experiments. Scale bar = 50 μm. Statistical analysis was performed one-way ANOVA with Tukey's (panel **B**) or Dunnett's (panel **E**&**F**) correction for multiple comparisons in GraphPad Prism Version 9.4.1 and 10.2.3.

The online version of this article includes the following source data and figure supplement(s) for figure 4:

**Source data 1.** Data points used to generate the graphs in *Figure 4E and F*.

**Source data 2.** Annotated immunoblots from *Figure 4G*.

**Source data 3.** Raw immunoblots from *Figure 4G*.

**Figure supplement 1.** Downregulated intracellular proteins in HDMECs following 24 hr exposure to 10 ng/mL mycolactone, according to subcellular location, presented as percentage of total.

**Figure supplement 2.** Impact of membrane anchor ΔG values on fold change in expression induced by mycolactone for all identified Type II membrane proteins.

**Figure supplement 3.** Cartoon representing mycolactone-downregulated steps of GAG enzymatic synthesis.

III digestion, dHS, disrupted surface HS expression was observed in mycolactone-exposed cells (14.11 ± 7.40% vs. DMSO solvent control 105.30 ± 9.79%, p=0.0006, *Figure 4F*). In addition, HS-containing proteoglycans were detected by immunoblot using the anti-dHS antibody. Heparinase III digestion revealed an abundance and diversity of heparan sulphate containing proteins present in untreated or DMSO-exposed HDMECs that decreased progressively with mycolactone exposure (*Figure 4G*, *Figure 4—source data 2*, *Figure 4—source data 3*). By immunofluorescence, HS forms a mesh-like network around and between cells in untreated and DMSO solvent controls (*Figure 4H*). However, in HDMECs exposed to mycolactone, or ZIF-80, the HS-positive network was disrupted within 20 hr (*Figure 4H*). Collectively, this data confirms that Sec61 inhibition by mycolactone profoundly impairs the ability of endothelial cells to synthesise GAG chains.

## Loss of galactosyltransferase II drives changes in endothelial cell morphology and monolayer permeability

We reasoned that mycolactone-dependant depletion of any enzyme involved in the early stages of GAG biosynthesis would, on its own, be sufficient to explain the loss of HS and CS. Therefore, we validated its effect on the GAG linker building enzyme galactosyltransferase II (B3GALT6) by immunofluorescence. Endothelial B3GALT6 colocalised with the Golgi marker GOLGB1/Giantin in a perinuclear region in untreated cells and was unchanged in those exposed to the solvent control (0.02% DMSO; *Figure 5A*). B3GALT6 expression levels remained normal in HDMECs exposed to mycolactone for 6 hr but a clear reduction was seen after 12 hr (*Figure 5A*). Similar findings were made with biological mycolactone (*Figure 5—figure supplement 1*). Notably, ZIF-80 reduced B3GALT6 expression in a similar manner (*Figure 5—figure supplement 2*).

In order to investigate whether loss of B3GALT6 was sufficient to induce the phenotypic changes we saw after mycolactone exposure, we knocked down B3GALT6 in HUVECs using siRNA. The reduction in B3GALT6 protein expression compared to cells transfected with non-targeting si-control RNA (*Figure 5B*) was comparable to that caused by mycolactone (~80%). B3GALT6 siRNA-treated cells

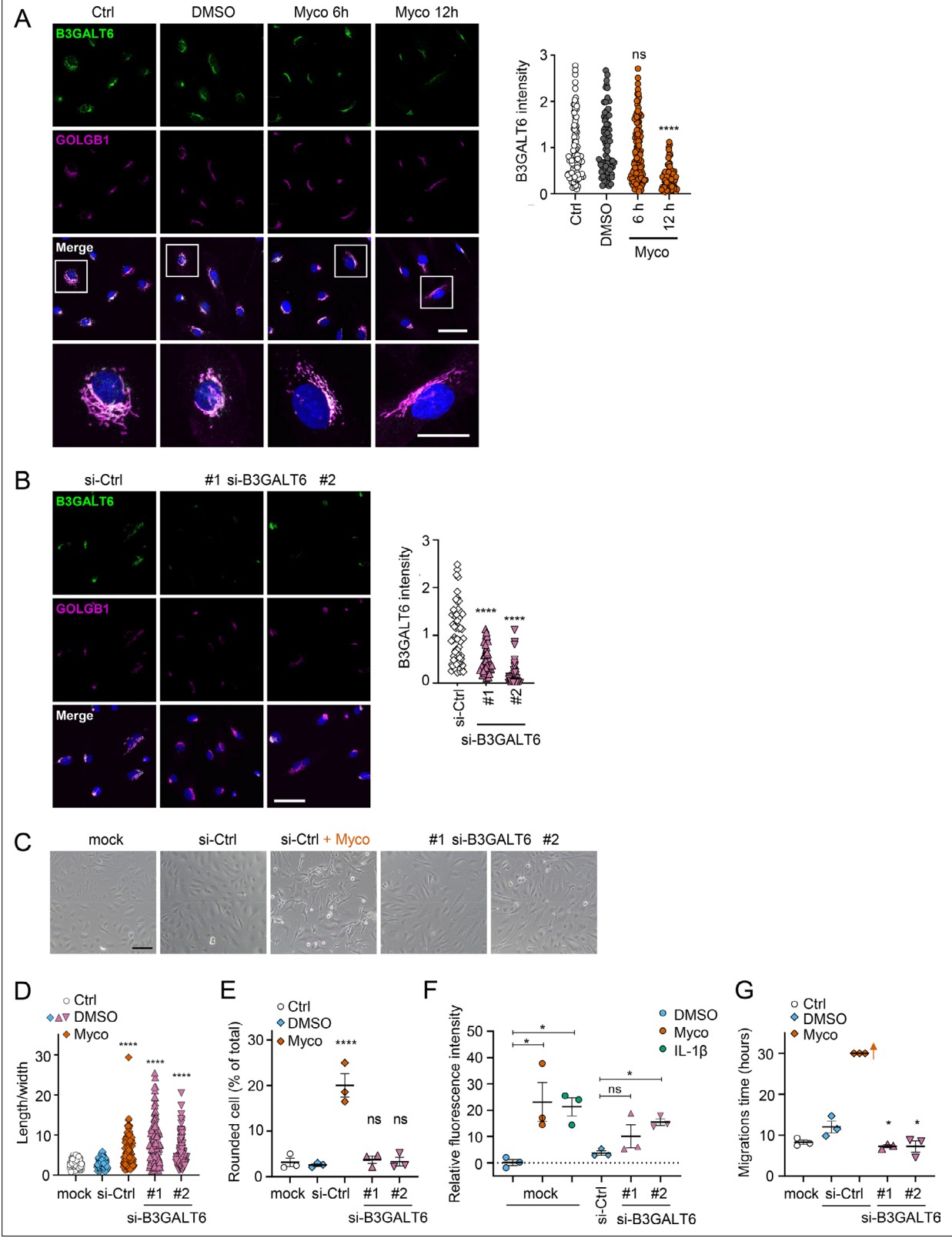

**Figure 5.** Loss of B3GALT6 affects endothelial cell morphology and monolayer permeability in primary human endothelial cells. (**A**) HDMECs exposed to 10 ng/mL mycolactone (Myco) or 0.02% DMSO for indicated times. (**B–G**) HUVECs transfected with si-B3GALT6 or si-negative control (Ctrl) oligos for 48 hr. (**A–B**) Cells were fixed, permeabilised and immunostained with anti-B3GALT6 and anti-GOLGB1 antibodies. B3GALT6 (green) and the Golgi apparatus (magenta) were visualised and nuclei stained with DAPI (blue). Scale bar = 50 µm (20 µm in the crop panels of **A**). Corrected total cell

*Figure 5 continued on next page*

*Figure 5 continued*

fluorescence of B3GALT6 in Golgi apparatus per cell was measured and presented as a value normalised to the mean value obtained from untreated control of each experiment. More than 30 cells per condition were measured per experiment. Images and quantification are representative of three independent experiments. (**C–E**) HUVECS exposed to 10 ng/mL mycolactone (Myco) or 0.02% DMSO for 24 hr one day post-transfection were imaged by an inverted microscope. (**D**) Length and width of each cell presented as a ratio. At least 100 cells were measured for each treatment. Values are representative of three independent experiments. (**E**) Rounded cell number per image presented as a % of total cell number per condition (values represent the mean ± SEM of three independent experiments). (**F**) Permeability of transfected HUVEC monolayers on inserts with 1 μm pores treated with 100 ng/mL IL-1β, 10 ng/mL mycolactone (Myco) or 0.02% DMSO for 24 hr was quantified. Fluorescence intensity of FITC-dextran in the receiver wells was measured and presented as a % where 100% is the value obtained from transwells lacking a cell monolayer, and 0% is untreated control wells (mean ± SEM of three independent experiments). (**G**) HUVECs were transfected with si-B3GALT6 or si-negative control (si-Ctrl) oligos. A scratch was introduced to the monolayer prior to the treatment (10 ng/mL mycolactone (Myco) or DMSO) and live cell imaging was performed with the zenCELL Owl incubator microscope every 15 min for 30 hr. Migration time in hours (hrs) to reform the monolayer is presented as mean ± SEM (n=3); wells with no visible monolayer at the end point were given a maximum value = 30. ns, not significant; *, p<0.05; ****, p<0.0001. Statistical analysis was performed using one-way ANOVA with Dunnett's correction for multiple comparisons in GraphPad Prism Version 9.4.1 and 10.2.3 (all analysed panels).

The online version of this article includes the following source data and figure supplement(s) for figure 5:

**Source data 1.** Data points used to generate the graphs in *Figure 5A, B, D, E, F and G*.

**Figure supplement 1.** HDMECs exposed to 10 ng/mL synthetic (Myco-Syn) or natural mycolactone (Myco-Bio) or 0.02% DMSO for 24 hr were fixed, permeabilised and immunostained with anti-B3GALT6 and anti-GOLGB1 antibodies.

**Figure supplement 2.** HDMECs exposed to 10 ng/mL of mycolactone (Myco), 0.02% DMSO, 20 nM ZIF-80 or untreated for 24 hr were fixed, permeabilised and immunostained with anti-B3GALT6 and anti-GOLGB1 antibodies and nuclei stained with DAPI.

demonstrated a similar elongated appearance (*Figure 5C*) and image analysis confirmed a significant increase in the ratio of cell length to width in HUVECs transfected with si-B3GALT6 RNA (*Figure 5D*). However, knockdown of B3GALT6 did not recapitulate the cell rounding phenotype (*Figure 5E*).

We next investigated the potential contribution of B3GALT6 loss to the previously observed mycolactone-induced increase in HDMEC and human dermal lymphatic endothelial cell monolayer permeability (*Hsieh et al., 2022*). Exposure of mock-transfected HUVEC monolayers to 10 ng/mL mycolactone for 24 hr increased permeability to 23.13 ± 7.38%, an effect comparable to 100 ng/mL IL-1β (21.30 ± 3.48%; *Figure 5E*). B3GALT6 knockdown in HUVECs also led to a rise in monolayer permeability (10.08 ± 4.37% and 15.47 ± 1.27% of the values seen in empty wells, p=0.2371 and 0.0367, for two different oligonucleotides, *Figure 5F*). Interestingly, B3GALT6 knockdown did not reduce the rate of HUVEC migration in scratch assays (*Figure 5G*); instead the cells exhibited a slightly increased healing rate compared to controls.

## Mycolactone rapidly depletes endothelial surface proteoglycans

Since loss of GAGs did not explain all the phenotypes observed, we considered the so-called core proteins to which GAGs synthesised in the Golgi are covalently linked to form the proteoglycans. These can be secretory, plasma membrane or GPI-anchored proteins, all of which require the Sec61 translocon for their biogenesis. Our proteome revealed that seven HS, CS, and/or DS-carrying proteoglycans were significantly down-regulated after 24 hr mycolactone exposure (*Figure 6A*).

Using flow cytometry, we validated the changes in abundance of three cell surface proteoglycans; perlecan (HSPG2; secreted, HS/CS), glypican-1 (GPC1; GPI-anchored, HS/CS) and biglycan (BGN; secreted, CS/DS). Syndecan-2, a membrane-bound protein for which only one unique peptide was found in the proteome, could not be detected by flow cytometry. The most profound effects were seen for perlecan and glypican-1 (detection at 10.8 ± 4.8% and 28.8 ± 9.0% of untreated control, *Figure 6B*), while biglycan was partly reduced (43.7 ± 6.8% of untreated control). As the turnover rate of HS proteoglycans is rapid $t_{1/2}$ = 3–4 hr in granulosa and 6.9 hr in macrophages (*Owens and Wagner, 1991*; *Yanagishita and Hascall, 1984*), we explored the rate of perlecan and glypican-1 loss at early time points in HUVECs. A~50% reduction in perlecan was evident after only 2 hr mycolactone treatment, reaching significance at 6 hr. Depletion of glypican-1 was slower, evident at 6 hr and reaching significance at 24 hr (*Figure 6—figure supplement 1*).

Immunofluorescence staining of HDMECs showed abundant perlecan staining in control cells, particularly around intercellular junctions, but the staining rapidly decreased in response to mycolactone, with reduced expression detectable after 8 hr (*Figure 5C*). HDMECs exposed to ZIF-80 for 8 hr displayed similarly limited perlecan-positive junctional staining (*Figure 6C*), and the depletion

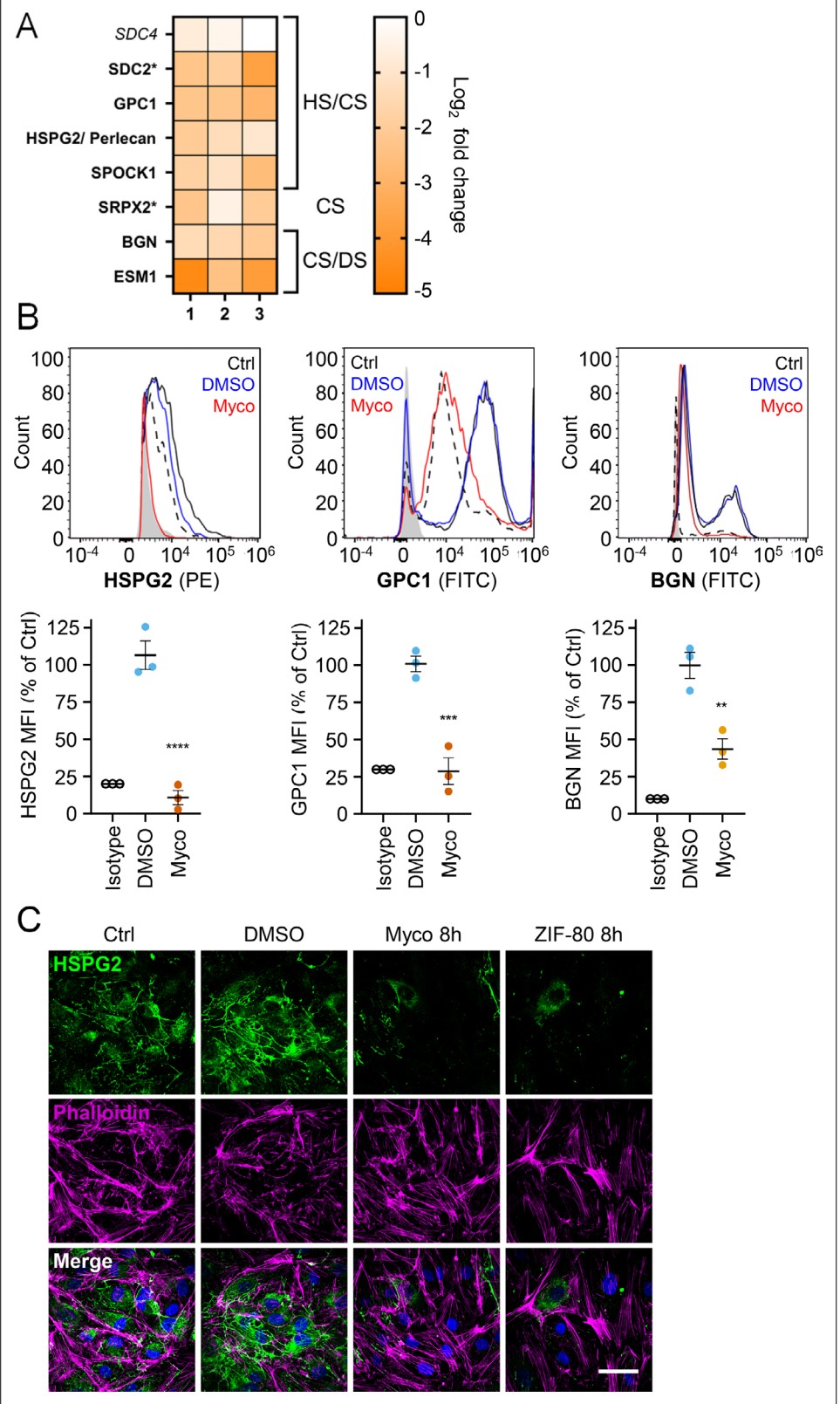

**Figure 6.** Mycolactone causes a rapid loss of multiple proteoglycans in primary human endothelial cells. HDMECs exposed to 10 ng/mL mycolactone (Myco), 0.02% DMSO or 20 nM ZIF-80 for 24 hr or indicated times. (**A**) Heatmap showing representative data for genes encoding proteoglycans. Dual-colour coding for log2 fold change in response to Myco is shown. Possible attached glycosaminoglycan chains such as heparan sulphate

*Figure 6 continued on next page*

*Figure 6 continued*

(HS), chondroitin sulphate (CS) or dermatan sulphate (DS) shown. Candidates with one unique peptide detected indicated with asterisks, significantly downregulated (p<0.05) or not (p≥0.05) in bold or *Italic,* respectively. (**B**) Cells were harvested for flow cytometry analysis. Histogram plots for single cell population of HSPG2, GPC1, and BGN. Unstained untreated cells, filled grey; isotype control of untreated cells, dashed black line. untreated cells stained with antibodies, black line; cells exposed to DMSO stained with antibodies, blue line; cells exposed to mycolactone stained with antibodies, red line. MFI is presented as a % of untreated control (mean ± SEM of three independent experiments). **, p<0.01; ***, p<0.001; ****, p<0.0001. (**C**) Cells were fixed and immunostained with anti-perlecan antibody (green), permeabilised and labelled with TRITC-conjugated phalloidin (magenta). Nuclei were stained with DAPI (blue). Images are representative of three independent experiments. Scale bar = 50 μm. Statistical analysis was performed using one-way ANOVA with Dunnett's correction for multiple comparisons in GraphPad Prism Version 9.4.1 and 10.2.3 (all analysed panels).

The online version of this article includes the following source data and figure supplement(s) for figure 6:

**Source data 1.** Data points used to generate the graphs in *Figure 6B*.

**Figure supplement 1.** HUVECs exposed to 10 ng/mL of mycolactone (Myco), 0.02% DMSO or remained untreated for indicated times.

**Figure supplement 2.** HDMECs exposed to 10 ng/mL synthetic (Syn) or natural mycolactone (Nat) or 0.02% DMSO for 24 hr.

**Figure supplement 3.** Confluent HDMECs under uniaxial shear stress for 24 hr were then exposed to 0.02% DMSO or 10 ng/ml mycolactone (Myco) for 48 hr under the same conditions then fixed and stained with anti-HSPG2 antibody (green) and counterstained with DAPI (blue).

was duplicated when biological purified mycolactone was compared to synthetic material (*Figure 6— figure supplement 2*), and also occurred when the endothelial cells were under shear stress (*Figure 6—figure supplement 3*). The parallel loss of GAGs and the proteoglycans that bear them means that the glycocalyx is severely disrupted by mycolactone.

## Mycolactone depletes endothelial basement membrane components and their ligands

Taken together, our results so far show that mycolactone profoundly depletes the endothelial glycocalyx, due to the loss of both GAG and proteoglycan biosynthesis following Sec61 inhibition. However, while loss of GAG production affected permeability, it had less impact on adhesion and migration. We therefore next focused on the downregulated proteins in our dataset with GO classifications linked to these processes. Numerous adhesion molecules and basement membrane components were downregulated by mycolactone, including nidogen 1 (NID1), laminins and collagens (*Figure 7A*). Although the abundance of major BM component collagen IV was not significantly influenced by mycolactone, perhaps indicating a slow turnover rate, several ER-localised and/or secreted enzymes involved in collagen biosynthesis, were reduced as previously reported in murine fibroblasts (*Gama et al., 2014*). Laminins are the other key constituent glycoproteins of the BM and important binding partners for endothelial cell integrins. Our proteomic data suggested multiple laminins are affected by mycolactone. Laminin α4 and α5 are both common to all types of vessel wall, but α4 has a slightly higher turnover rate (*Sixt et al., 2001*). By immunofluorescence staining, laminin α4 was seen in perinuclear regions within cells and in the network bridging intercellular junctions between endothelial cells in untreated and DMSO control HDMEC (*Figure 7B*). After 16 hr of exposure to mycolactone, the perinuclear staining was absent and the laminin-positive network between cells had become disconnected (*Figure 7B*). Similar findings were made in biological purified mycolactone was compared to synthetic material (*Figure 7—figure supplement 1*). Loss of laminin α4 staining was also observed in endothelial cells exposed to mycolactone under shear stress (Fig *Figure 7—figure supplement 2*). The same striking decrease was also seen in HDMECs exposed to ZIF-80 (*Figure 7B*).

The effect of mycolactone on the abundance of the laminin binding integrin β subunits β1 and β4 and laminin α5 in HDMEC were determined by flow cytometry (*Figure 7C*). After 24 hr, they were reduced to 45.0 ± 6.2%, 27.3 ± 7.7% and 15.6 ± 5.4% respectively of control levels (*Figure 7C*). In addition, the loss of expression of the basement membrane component fibronectin and cell surface integrin α5 were validated using immunoblot analysis; fibronectin levels decreased very rapidly showing >75% depletion after 4 hr exposure to mycolactone (p<0.01; *Figure 7—figure supplement*

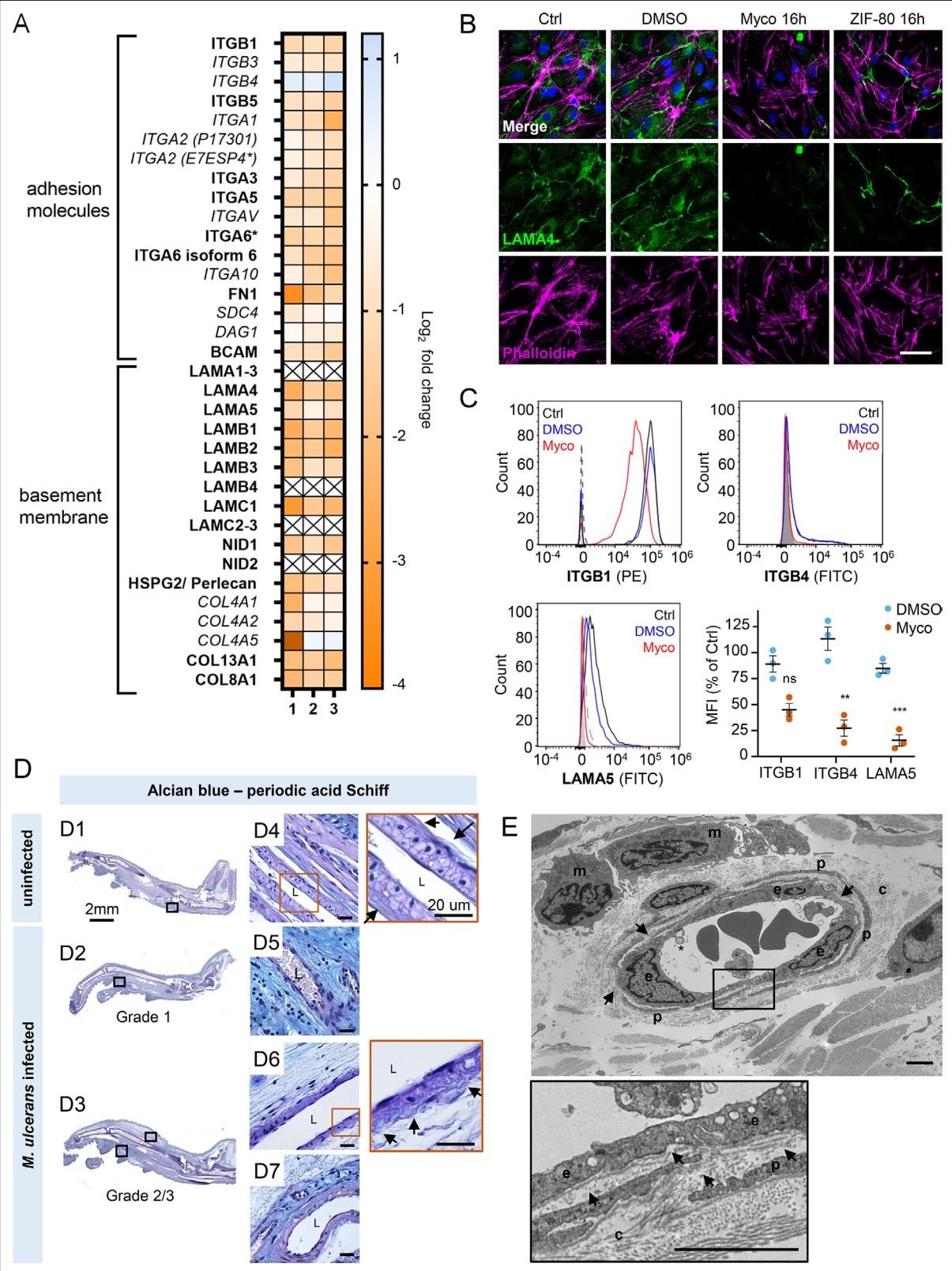

**Figure 7.** Mycolactone depletes primary human endothelial adhesion molecules and basement membrane proteins, and damages the basement membrane during mouse infection. (**A–C**) HDMECs exposed to 10 ng/mL mycolactone (Myco), 0.02% DMSO or 20 nM ZIF-80 for 24 hr or indicated times. (**A**) Heatmap showing representative data for genes encoding junctional or adhesion molecules, basement membrane components and proteins involved in platelet adhesion. Dual-colour coding for log2 fold change in response to Myco is shown. Candidate with one unique peptide detected is

*Figure 7 continued on next page*

*Figure 7 continued*

indicated with asterisks, significantly downregulated (p<0.05) or not (p≥0.05) in bold or *Italic,* respectively. (**B**) HDMEC were fixed and immunostained with anti-laminin α4 antibody (green), permeabilised and labelled with TRITC-conjugated phalloidin (magenta). Nuclei were stained with DAPI (blue). Images are representative of two independent experiments. Scale bar = 50 µm. (**C**) HDMEC were harvested for flow cytometry analysis. Histogram plots for single cell population of integrin β1, integrin β4, and laminin α5. Unstained, untreated cells, filled grey; isotype control of untreated cells, dashed black line. untreated cells stained with antibodies, black line; cells exposed to DMSO stained with antibodies, blue line; cells exposed to Myco stained with antibodies, red line. MFI is presented as a % of untreated control (mean ± SEM of three independent experiments). Statistical analysis was performed using one-way ANOVA with Dunnett's correction for multiple comparisons in GraphPad Prism Version 9.4.1 and 10.2.3; ns, not significant; **, p<0.01; ***, p<0.001. (**D–E**) C57BL/6 J mice were infected with *M. ulcerans* bacteria using the footpad model (**D**) Alcian blue-periodic acid Schiff stain of mice that received vehicle control (PBS) (**D1, D4**) or intradermal injection of 1*10$^5$ colony forming units *M. ulcerans* at 21 (Grade 1; **D2, D5**) or 28 days (Grade 2/3; D3, D6-7) post-infection. Neutral glycans are indicated by purple staining and acidic glycans by light blue Blood vessel lumens are indicated by an "L". Scale bars in D1-3: 2 mm; all others: 20 µm. (**E**) Representative transmission electron microscopy scan of grade 1 *M. ulcerans* infected murine footpad. The ultrathin section of glabrous skin shows the pericapillary interstitium slightly expanded, collagen (**c**) fibrils variably disaggregated and infiltrating macrophages (**m**). The endothelial cells (**e**) are reactive and exhibit cytoplasmic projections (*) whilst the basement membrane is multifocally disrupted (arrows). A thin layer of pericytes (**p**) is variably expanded by oedema. Scale bar: 2 µm.

The online version of this article includes the following source data and figure supplement(s) for figure 7:

**Source data 1.** Data points used to generate the graphs in *Figure 7C*.

**Source data 2.** Annotated immunoblots from *Figure 7—figure supplement 3*.

**Source data 3.** Raw immunoblots from *Figure 7—figure supplement 3*.

**Source data 4.** Annotated immunoblots from *Figure 7—figure supplement 4*.

**Source data 5.** Raw immunoblots from *Figure 7—figure supplement 4*.

**Figure supplement 1.** HDMECs exposed to 10 ng/mL synthetic (Myco-Syn) or natural mycolactone (Myco-Bio) or 0.02% DMSO for 24 hrs.

**Figure supplement 2.** Confluent HDMECs under uniaxial shear stress for 24 hr were then exposed to 0.02% DMSO or 10 ng/ml mycolactone (Myco) for 48 hr under the same conditions, then fixed and stained with anti-LAMA4 antibody (green) and counterstained with DAPI (blue).

**Figure supplement 3.** HDMECs exposed to 10 ng/mL of mycolactone (Myco), 0.02% DMSO or remained untreated for indicated times.

**Figure supplement 4.** HDMECs exposed to 10 ng/mL of mycolactone (Myco), 0.02% DMSO or remained untreated for indicated times.

*3*, *Figure 7—source data 2*, *Figure 7—source data 3*) whilst the level of integrin α5 decreased more slowly, reaching ~50% of control levels at 24 hr (p<0.01; *Figure 7—figure supplement 4*, *Figure 7—source data 4*, *Figure 7—source data 5*).

To determine whether the basement membrane was disrupted in vivo, we stained the tissue sections from *M. ulcerans*-infected mice with the alcian blue-periodic acid Schiff (AB-PAS) method. In mouse feet receiving the vehicle control (*Figure 7D1*), the dermis contained neutral glycans (purple staining) and the vasculature displayed an intact vessel basement membrane (*Figure 7D4*, insert). At early stages of infection (Grade 1; *Figure 7D2*), immune cell infiltration could be seen in these regions in proximity to mycobacterial clusters (*Figure 2—figure supplement 1*) and the surrounding dermal tissue had become more acidic (blue staining; *Figure 7D5*). At later stages of infection, when the metatarsal area was more swollen (*Figure 7D3*) and the dermis showed marked oedema and the fibrous architecture was disrupted (*Figure 7D6–7*), there was an overall reduction in the intensity of staining around the vasculature and the vessel basement membranes were irregular (*Figure 7D6*, insert).

To confirm the impact of infection on the basement membrane we used transmission electron microscopy (*Figure 7E*) to characterize the early vascular changes in the podal dermis of mice inoculated with *M. ulcerans*. Multiple transverse, oblique and longitudinal sections of arterioles, venules and lymphatics were examined. In Grade 1, the interstitium surrounding the capillary was expanded by electron-lucent granular material, pericyte processes were often separated and collagen fibrils close to the blood vessels increasingly disaggregated. The endothelial cells exhibited varying degrees of swelling and vacuolation with an irregular luminal surface including cytoplasmic undulation and projections. The subendothelial basement membrane appeared multifocally disrupted, expanded or discontinuous (*Figure 7E*, arrows). Taken together the data shows that even at early stages of infection, the endothelial basement membrane is compromised and the loss of constituent proteins caused by mycolactone is likely a major factor in these changes.

## Exogenous laminin-α5 ameliorates mycolactone-driven cell detachment and impaired migration

Since laminins are secreted proteins, which are then deposited to form cell-associated extracellular matrix, we wondered whether exogenous provision of these molecules might protect mycolactone-exposed cells. We therefore coated tissue culture plates with laminin-111,–411 or –511, complexes that contain laminin β1γ1 in combination with laminins α1, α4, or α5 respectively. As expected (*Di Russo et al., 2017*), primary HDMECs efficiently re-attached to laminin-511-coated culture vessels, with very little reattachment to uncoated vessels (p=0.0020, *Figure 8—figure supplement 1*). Re-attachment to laminin-411 or the non-endothelial specific laminin-111 was also observed albeit to a lesser extent (p=0.1226 and 0.3365 compared to the uncoated wells, respectively). We then quantified the re-attachment of endothelial cells that had been pre-exposed to mycolactone for 24 hr compared to controls (*Figure 8A*). Remarkably, mycolactone-exposed cells re-adhered to specifically to laminin-511- (but not 411- or 111-) coated vessels with the same efficiency as controls (*Figure 8A*).

We then investigated whether exogenous laminin-511 could ameliorate the cell rounding, attachment or migration phenotypes observed in response to mycolactone using time-lapse imaging of HDMECs. On uncoated wells, mycolactone caused the expected phenotypic changes (*Figure 8B–C*), and remarkably, exogenous laminin α5 significantly reduced mycolactone-driven cell rounding, even after 48 hr (7.7 ± 1.5% vs 17.6 ± 2.4%, p=0.0153, *Figure 8B*). Similarly, while the relative number of attached cells did not increase steadily with time as for the DMSO control (*Figure 8C*), laminin-511 coating prevented the decrease in attached cells seen between 36 and 48 hr in uncoated wells (p=0.0123). These effects were absent in laminin-411 and –111-coated wells (*Figure 8—figure supplement 2*). Laminin coating did not impact HDMEC survival in the presence or absence of mycolactone at 48 hr (*Figure 8—figure supplement 3*), although as mentioned before, cell death due to mycolactone is minimal prior to 72 hr (*Ogbechi et al., 2015*).

For migration, we performed a scratch assay on HUVECs in wells coated or not with laminins prior to mycolactone exposure. Monitoring cell migration using time-lapse imaging revealed that control HUVECs took less than 16 hr to close a 600–800 µm gap (*Figure 8—video 1*). By contrast, the leading edge of wounded HUVEC monolayers exposed to mycolactone gradually stopped migrating into the cell-free region after ~7 hr; at this point the cells began to migrate randomly before undergoing the previously described morphological changes (*Figure 8—video 2*). However, strikingly, in HUVEC monolayers plated onto laminin-511, cells continued to migrate into the gap in the presence of mycolactone, with a leading edge still evident after 16 hr (*Figure 8—video 3*). Cell counts per unit scratch area at 8 and 16 hr showed that cells plated onto laminin-511 were able to migrate back at a rate comparable to that seen in monolayers exposed to DMSO in uncoated wells (*Figure 8D*, p=0.286). We did not see these same effects on laminin-411 and –111-coated wells (*Figure 8—video 4* and *Figure 8—video 5*) where migration rates remained significantly lower than the control (p=0.0054 and 0.0003, respectively, *Figure 8—figure supplement 4*).

This ability of laminin α5 to reverse or diminish the impact of mycolactone on endothelial cell adhesion, morphology and migration highlights the contribution of the loss of basement membrane proteins to the phenotypic changes induced by mycolactone and presents an unanticipated potential for use in wound care in Buruli ulcer skin lesions, although such therapies are currently in their infancy (*Iorio et al., 2015*).

## Discussion

Until recently, the pathogenesis of BU was thought to rely on two factors; immunosuppression due to the action of mycolactone on innate and adaptive immunity, and direct cytotoxic action of mycolactone on the cells present within the subcutis leading to cell death and necrosis. Our findings provide further evidence supporting a third and vital pathway to tissue necrosis; the induction of endothelial dysfunction that drives an indirect mechanism leading to tissue necrosis via the breakdown of vessel integrity and fibrin-driven ischemia within tissue.

The current work reaffirms the critical role that Sec61 inhibition plays in the virulence mechanism of mycolactone. In this post-transcriptional, co-translational mechanism responsible for changes in protein abundance, proteins are made in the wrong cellular compartment (the cytoplasm) and degraded by the ubiquitin-proteasome system or removed by autophagy (*Hall et al., 2014*; *Hall*

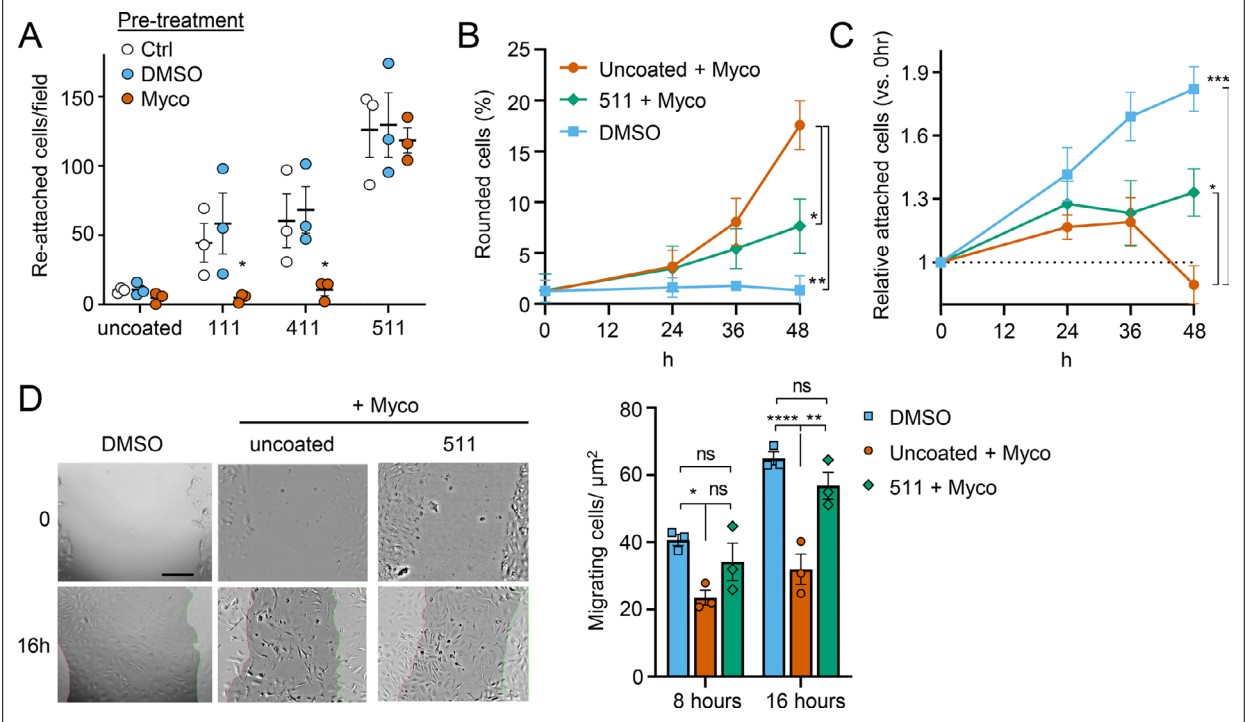

**Figure 8.** Laminin α5 ameliorates mycolactone-driven primary human endothelial cell detachment and impaired migration. Endothelial cells exposed to 10 ng/mL mycolactone (Myco) or 0.02% DMSO for 24 hr or indicated times. (**A**) Treated HDMECs were harvested and re-seeded to laminin-511, 411, 111 or uncoated plates. After an hour, unbound cells were washed away and attached cells were imaged and cell numbers per field are presented as mean ± SEM of three independent experiments. (**B–D**) Endothelial cells seeded onto laminin-511 or uncoated plates were exposed to mycolactone (Myco) or DMSO. (**B–C**) HDMECs were imaged every 30 min over 48 hr. Rounded or attached cells per condition were counted at 0, 24, 36, 48 hr. Data are presented as a % of total cell number of each condition (**B**) or normalised to the attached cell number counted at 0 hr (**C**) (mean ± SEM of three independent experiments). (**D**) A scratch was introduced to a HUVEC monolayer prior to treatment. The wounded area was imaged every 15 min for 24 h. Scale bar: 200 μm. Cells migrating into the original scratch area were counted at 0, 8, and 16 hr. Data are presented as cell count per scratch area (mean ± SEM of three independent experiments). ns, not significant; *, p<0.05; **, p<0.01; ***, p<0.001; ****, p<0.0001. Statistical analysis was performed in GraphPad Prism Version 9.4.1 and 10.2.3 using two-way ANOVA with Tukey's correction for multiple comparisons (all analysed panels; panels **B&C** also included the Geisser Greenhouse correction for sphericity).

The online version of this article includes the following video, source data, and figure supplement(s) for figure 8:

**Source data 1.** Data points used to generate the graphs in *Figure 8A, B, C and D*.

**Figure supplement 1.** HDMECs were harvested and layered to laminin-511, 411, 111 or uncoated wells for one hour.

**Figure supplement 2.** HDMEC seeded onto laminin-411, 111 or uncoated plates were exposed to mycolactone (Myco) or DMSO and imaged every 30 min over 48 hr.

**Figure supplement 3.** HDMECs seeded onto different laminin isoforms were untreated or exposed to 0.02% DMSO or 10 ng/mL mycolactone (Myco) for 48 hr.

**Figure supplement 4.** A scratch was introduced to a HUVEC monolayer prior to treatment.

**Figure 8—video 1.** Scratch repair in HUVECs exposed to 0.02% DMSO.

https://elifesciences.org/articles/86931/figures#fig8video1

**Figure 8—video 2.** Scratch repair in HUVECs exposed to 10 ng/mL mycolactone.

https://elifesciences.org/articles/86931/figures#fig8video2

**Figure 8—video 3.** HUVECs were seeded on to laminin-511 coated well.

https://elifesciences.org/articles/86931/figures#fig8video3

**Figure 8—video 4.** HUVECs were seeded on to laminin-411 coated well.

https://elifesciences.org/articles/86931/figures#fig8video4

**Figure 8—video 5.** HUVECs were seeded on to laminin-111 coated well.

https://elifesciences.org/articles/86931/figures#fig8video5

*et al., 2022*). During the current studies we tried, without success, to express examples of our library of SEC61A1 mutants that confer resistance to mycolactone (*Gérard et al., 2020*; *Ogbechi et al., 2018*) in primary endothelial cells. This suggests that the endothelium is particularly sensitive to functional perturbation of the Sec61 translocon and perhaps explains why these cells are so exquisitely sensitive to the compound. As an alternative approach, we tested two analogues of the structurally unrelated Sec61 inhibitor Ipomoeassin F that was first isolated as a natural product of the 'Morning Glory' flower (*Zong et al., 2019*). Across multiple readouts, this induced comparable phenotypes to mycolactone, in the same time frame, including changes in morphology, loss of GAGs, matrisome proteins required for their synthesis, proteoglycan core proteins and basement membrane proteins. Hence, we are confident that the primary target of mycolactone in endothelial cells is Sec61, as it has already been shown in immune cells (*Baron et al., 2016*; *Hall et al., 2014*; *Grotzke et al., 2017*), fibroblasts (*Hall et al., 2022*) and epithelial cells (*Ogbechi et al., 2018*). Since epithelial cells showed similar effects on migration to those we report here, these effects probably depend on its action on the Sec61 translocon rather than other previously proposed mechanisms such as WASP activation (*Guenin-Macé et al., 2013*).

Although the Sec61 translocon is thought to be required for the biosynthesis of approximately 30% of the proteome, mycolactone only inhibits production of specific subsets of proteins that traffic through the ER. As their depletion rate depends on protein turnover, inhibitory effects cannot be predicted a priori. Therefore, we performed quantitative proteomic analysis of total membrane fractions of primary endothelial cells to identify as many of the targets of mycolactone as possible, since 'whole cell' approaches can bias against membrane proteins, particularly insoluble ones. Indeed, this approach was successful, more than doubling the number of detected proteins classified as membrane, secretory, ER/ERGIC, Golgi or endolysosomal compared to previous studies (*Morel et al., 2018*). Significant depletion of our previously discovered protein targets including loss of CDH5, TIE-1, TEK, JAM-C, CD31, vWF, TFPI, and TM (*Ogbechi et al., 2015*; *Hsieh et al., 2022*), and induction of SQSTM1/p62 (*Hall et al., 2022*) validates this data set. The pattern of protein topologies affected by mycolactone reflected that seen in in vitro translocation assays and whole cell proteome analysis (*Baron et al., 2016*; *McKenna et al., 2017*; *Morel et al., 2018*; *Gama et al., 2014*; *McKenna et al., 2016*), further supporting mycolactone selectivity towards secreted, Type I and Type II single pass membrane proteins, with few multi-pass proteins and no Type III or tail- anchored membrane proteins showing any reduction in expression.

GO analysis confirmed induction of cytoplasmic/oxidative stress responses (*Ogbechi et al., 2018*; *Förster et al., 2020*; *Kwaffo et al., 2021*) amongst >220 up-regulated proteins in mycolactone-exposed endothelial cells. However, in this work, we focussed on the >480 down-regulated proteins, which represented a striking inadequacy in components of glycoprotein biosynthesis and metabolism and ECM organization, many of which we have validated individually. Taking together the cellular compartment analysis and our understanding of mycolactone's cellular target, we were able to correctly hypothesise that the effects on endothelial cell function were exacerbated by loss of Golgi-localised Type II transmembrane enzymes involved in GAG (CS, DS, and HS) production. Up to now, the impact of mycolactone on Golgi function has been underappreciated, but the wider ranging effect on protein glycosylation may explain why mycolactone has such a strong effect on glycosylated protein production irrespective of topology (*Hall et al., 2014*). It also suggests that the effects of mycolactone may be even more far-reaching than expected, as even molecules resistant to the Sec61 blockade at the protein level may be functionally affected due to the loss of glycosylation.

Since GAG biosynthesis is a sequential process, and the endothelial glycocalyx is essential to maintain monolayer permeability (*Harding et al., 2019*), we reasoned that loss of one of the GAG linker-building enzymes common to CS, DS, and HS could be sufficient to explain the mycolactone-induced phenotype. This was supported by siRNA-mediated knockdown of B3GALT6, which transfers galactose to substrates such as galactose-beta-1,4-xylose, that is the third step in this process. B3GALT6 knockdown phenocopied the elongated appearance seen in primary endothelial cells exposed to mycolactone. The intermediate phenotype seen in some experiments suggests that depletion of junctional molecules by mycolactone (*Hsieh et al., 2022*), also plays an important contributory role. In the context of BU, it is interesting to note that children born with 'linkeropathies', who have a reduced ability to synthesise GAG linker regions (*Colman et al., 2019*; *Mihalic Mosher et al., 2019*), display phenotypes such as skin fragility and delayed wound healing (*Malfait et al., 2013*) that are similar to

antibiotic-treated *M. ulcerans* infections. As well as increasing permeability, the loss of the glycocalyx could exacerbate the inhibition of leukocyte homing caused by mycolactone (*Guenin-Macé et al., 2011*). Notably other viral and bacterial pathogens promote colonisation by degrading the endothelial glycocalyx (*Puerta-Guardo et al., 2016*; *Chiu et al., 2021*; *Tang et al., 2021*); however, here the mechanism is via inducing the production of heparanase and other proteinases.

Importantly, it is not only the GAGs of the apical glycocalyx that are depleted by mycolactone. Many proteoglycan core proteins are also lost. The secretory protein perlecan is notable for being a component of the glycocalyx as well as the BM and was profoundly and rapidly lost from the surface of primary endothelial cells following mycolactone exposure. Other BM components, particularly laminins and their cellular receptors, were also found to be depleted. Excitingly, providing an exogenous coating of laminin α5-containing laminin-511 complex to tissue culture wells protected endothelial cells from mycolactone-driven changes, improving adhesion, and reversing the migration defect. We have not been able to ascribe this to the retention of a specific adhesion molecule, and instead postulate that rescue could be via residual expression of a wide variety of laminin α5 receptors. This is supported by previous work showing that laminin α5 is more promiscuous than laminin α4 (*Di Russo et al., 2017*).

Adequate adhesion to the BM is critical for endothelial cell proliferation, migration, morphogenesis and survival (*Davis and Senger, 2005*). Furthermore, loss of perlecan and laminin α4, or reduced binding to fibronectin, disturbs the structural integrity and maturation of microvessels (*Thyboll et al., 2002*; *Gustafsson et al., 2013*; *Nicosia et al., 1993*), Finally, laminin α5 not only guides tissue patterning (*Spenlé et al., 2013*) and development (*Miner et al., 1998*) but also maintains vascular homeostasis by stabilising endothelial cell tight junctions (*Song et al., 2017*). Therefore, it is perhaps not surprising that we found the BM to be disturbed in *M. ulcerans* infected footpads. Moreover, this was seen in more advanced infections where fibrin deposition was also present within tissue, due to disturbance of the boundary between damaged vessels and dermal connective tissue. It is possible that these effects are exacerbated by IL-1β in vivo; this Sec61-independent pro-inflammatory cytokine has been shown to be induced in macrophages by mycolactone and *M. ulcerans* (*Foulon et al., 2020*; *Hall et al., 2021*) and is known to have profound effects on endothelial cell function, including the downregulation of anticoagulant and junctional proteins, induction of vascular permeability and upregulation of BM degrading proteinases (*Cottam et al., 1996*). There is considerable overlap in the endothelial cell responses to IL-1β and mycolactone, although the former's effects are mediated predominantly at the transcriptional level. An additive effect of mycolactone has been shown for some of these phenotypes in vitro (*Ogbechi et al., 2015*; *Hsieh et al., 2022*), although the in vivo situation is likely more complex (*Hall et al., 2021*).

In summary, this study identifies loss/disruption of the endothelial glycocalyx and BM as a critical molecular process in the pathogenesis of Buruli ulcer. These effects were the same with mycolactone extracted from *M. ulcerans* bacteria and chemically synthesised material, supporting the physiological relevance of our findings. Since these changes occur prior to mycolactone-driven apoptosis (*Ogbechi et al., 2018*; *Bieri et al., 2017*; *Ogbechi et al., 2015*), they provide further support for our working model whereby mycolactone builds a hyper-coagulative environment alongside disruption of the endothelial monolayer and BM integrity. We propose that this leads to leakage of high molecular weight plasma proteins into the connective tissue where they activate the coagulation cascade leading to fibrin deposition and tissue ischemia. The detection of extravascular fibrinogen at early stages of infection prior to widespread tissue damage and necrosis provides further evidence that endothelial dysfunction could be a driver of disease progression. Rethinking of BU as a vascular disease may ultimately lead to improved therapies that support better wound healing, alongside antibiotic treatment. However, it should be remembered that tissue repair requires a controlled progression through a series of different stages *Barker and Engler, 2017*; following injury, under normal circumstances, platelet accumulation in a fibrin and fibronectin rich matrix is followed by an inflammation phase (*Clark et al., 1982*). Therefore, ameliorating the coagulative features with anticoagulants alongside the standard antimycobacterial drugs may be of most value in the initial stages of treatment, while bioactive dressings containing laminin-derived peptides might be more useful to promote healing at later stages. In this context, laminin-derived bioactive peptides have recently been proposed as a treatment for defective tissue repair (*Iorio et al., 2015*) and indeed, accelerate re-epithelialisation in wounds of diabetic animals (*Zhu et al., 2018*; *Ishihara et al., 2018*), suggesting this novel approach

may be an effective complement to current therapies and could alleviate the long wound healing times experienced by BU patients.

# Materials and methods

## Key resources table

| Reagent type (species) or resource | Designation | Source or reference | Identifiers | Additional information |
|---|---|---|---|---|
| Strain, strain background (*Mus musculus*, female) | C57BL/6 J | Charles River | RRID:MGI:3028467 | |
| Strain, strain background (*M. ulcerans*) | Mu_1082 | Richard Phillips, KCCR, Ghana | | |
| Cell line (Human primary cells) | HUVEC | PromoCell | C-12200 | Single Donor |
| Cell line (Human primary cells) | HDMEC | PromoCell | C-12210 | Juvenile, Single Donor, male |
| Antibody | Anti-fibrinogen (rabbit polyclonal) | Agilent DAKO | A0080 RRID:AB_2894406 | IHC: (1:3000) |
| Antibody | Biotinylated anti-rabbit IgG (horse polyclonal) | Vector laboratories | BP-100–50 RRID:AB_3661924 | IHC (1:50) |
| Antibody | Anti-human Δ-HS (mouse monoclonal F69-3G10) | AMSBIO | 370260 S RRID:AB_10892311 | FACS (1:200) WB (1:1000) |
| Antibody | Anti-chondroitin sulphate (mouse monoclonal CS56) | Merck | C8035 RRID:AB_476879 | FACS (1:200) |
| Antibody | Anti-perlecan (mouse monoclonal 7B5) | Thermo Fisher Scientific | 13–4400 RRID:AB_86311 | FACS (1:200) IFA (1:500) |
| Antibody | Anti-glypican-1 (goat polyclonal) | Novus Biologicals | AF4519 RRID:AB_2232505 | FACS (2.5 µg/$10^6$ cells) |
| Antibody | Anti-integrin β4/CD104 (rat monoclonal 439-9B) | eBioscience | 14-1049-82 RRID:AB_1210460 | FACS (1:100) |
| Antibody | Anti-integrin β1/CD29 (mouse monoclonal P4C10) | Novus Biologicals | NBP2-36561 RRID:AB_3295906 | FACS (1:200) |
| Antibody | Anti-syndecan-2 (rat monoclonal 305515) | Novus Biologicals | MAB2965 RRID:AB_2182871 | FACS (0.25 µg/$10^6$ cells) |
| Antibody | Anti-biglycan (goat polyclonal) | Novus Biologicals | AF2667 RRID:AB_2065204 | FACS (0.1 µg/$10^6$ cells) |
| Antibody | Anti-laminin α5 (mouse monoclonal CL3118) | Novus Biologicals | NBP2-42391 RRID:AB_3306362 | FACS (1:200) |
| Antibody | Anti-laminin α4 (sheep polyclonal) | Biotechne | AF7340 RRID:AB_3644426 | IFA (1:200) |
| Antibody | Isotype control (mouse monoclonal IgG1κ P3.6.2.8.1) | Thermo Fisher Scientific (Invitrogen) | 14-4714-81 RRID:AB_470110 | FACS (1:100) |
| Antibody | Isotype control (mouse monoclonal IgG2b) | Thermo Fisher Scientific | MG2B00 RRID:AB_2921189 | As appropriate (same as test Ab) |
| Antibody | Isotype control (polyclonal goat IgG) | R&D Systems | AB-108-C RRID:AB_354267 | As appropriate (same as test Ab) |
| Antibody | Isotype control rat monoclonal IgG2bκ (eB149/10H5) | Thermo Fisher Scientific | 14-4031-81 RRID:AB_470098 | As appropriate (same as test Ab) |
| Antibody | Isotype control (mouse monoclonal IgM, clone PFR-03) | Thermo Fisher Scientific | MA1-10438 RRID:AB_2536806 | As appropriate (same as test Ab) |
| Antibody | PE-F(ab')2-anti-mouse IgG (rat polyclonal) | Thermo Fisher Scientific | 12-4010-82 RRID:AB_11063706 | As appropriate |
| Antibody | FITC-anti-goat IgG (donkey polyclonal) | Thermo Fisher Scientific | A16000 RRID:AB_2534674 | As appropriate |

*Continued on next page*

*Continued*

| Reagent type (species) or resource | Designation | Source or reference | Identifiers | Additional information |
|---|---|---|---|---|
| Antibody | FITC-anti-rat IgG (goat polyclonal) | Thermo Fisher Scientific | 31629 RRID:AB_228240 | As appropriate |
| Antibody | Anti-fibronectin (rabbit polyclonal) | Merck Millipore | AB1945 RRID:AB_2231910 | WB (1:1000) |
| Antibody | Anti-integrin α5 (mouse monoclonal IgG1κ A-11) | Santa Cruz Biotechnology | sc-166665 RRID:AB_2280538 | WB (1:1000) |
| Antibody | HRP-Anti-rabbit IgG (donkey polyclonal) | GE Healthcare | NA934V RRID:AB_2722659 | WB (1:5000) |
| Antibody | HRP-Anti-mouse IgG (sheep polyclonal) | GE Healthcare | NA931V RRID:AB_772210 | WB (1:5000) |
| Antibody | Anti-B3GALT6 (mouse polyclonal) | Bio-Techne | H00126792-B01P RRID:AB_3146818 | IF (1:100) |
| Antibody | Anti-giantin (rabbit polyclonal) | Abcam | Ab80864 RRID:AB_10670397 | IF (1:200) |
| Antibody | Alexa Fluor 594 anti-rabbit IgG (goat polyclonal) | Thermo Fisher Scientific | A11012 RRID:AB_2534079 | IF (1:400) |
| Antibody | Alexa Fluor 488 anti-mouse IgG (donkey polyclonal) | Thermo Fisher Scientific | A21202 RRID:AB_141607 | IF (1:400) |
| Antibody | Alexa Fluor 488 anti-sheep IgG (donkey polyclonal) | Thermo Fisher Scientific | A11015 RRID:AB_2534082 | IF (1:500) |
| Antibody | Alexa Fluor 647 anti-mouse IgG (goat polyclonal) | Thermo Fisher Scientific | A21235 RRID:AB_2535804 | IF (1:400) |
| Antibody | Alexa Fluor 647 anti-rabbit IgG (goat polyclonal) | Thermo Fisher Scientific | A21244 RRID:AB_2535812 | IF (1:400) |
| Sequence-based reagent | Silencer SiRNA negative control No 1 | ThermoFisher Scientific | AM4611 | 50 nM |
| Sequence-based reagent | Human B3GALT6 siRNA | ThermoFisher Scientific | Assay ID#112321 | 50 nM |
| Sequence-based reagent | Human B3GALT6 siRNA | ThermoFisher Scientific | Assay ID#112322 | 50 nM |
| Peptide, recombinant protein | Human Laminin-111 | BioLamina | LN111-02 | 5 µg/ml |
| Peptide, recombinant protein | Human Laminin-411 | BioLamina | LN411-02 | 5 µg/ml |
| Peptide, recombinant protein | Human Laminin-511 | BioLamina | LN511-0202 | 5 µg/ml |
| Peptide, recombinant protein | Human IL-1β | Gibco | PHC0813 | 100 ng/ml |
| Commercial assay or kit | ImmPACT NovaRED peroxidase substrate kit | Vector Laboratories | SK4805 | |
| Commercial assay or kit | VECTASTAIN Elite ABC kit | Vector Laboratories | PK6100 | |
| Chemical compound, drug | Dimethyl sulfoxide HYBRI-MAX | Merck | D2650 | |
| Chemical compound, drug | Synthetic Mycolactone | Prof Yoshito Kishi, Harvard University | CAS: 222050-77-3 | |
| Chemical compound, drug | Biological Mycolactone | Dr Estelle Marion, INSERM | CAS: 222050-77-3 | |
| Chemical compound, drug | Ipomoeassin F | Prof Wei Shi, University of Arkansas | CHEMBL4163767 | |
| Chemical compound, drug | ZIF-80 | Prof Wei Shi, University of Arkansas | None | |
| Software, algorithm | Image J (v1.52n) | Fiji | RRID:SCR_002285 | |
| Software, algorithm | Zencell-owl software (version 3.3) | innoME GmbH | | |
| Software, algorithm | Prism Version 9.4.1 and 10.2.3 | GraphPad | RRID:SCR_002798 | |
| Software, algorithm | ΔG prediction server v1.0 | dgpred.cbr.su.se | | |

*Continued on next page*

*Continued*

| Reagent type (species) or resource | Designation | Source or reference | Identifiers | Additional information |
|---|---|---|---|---|
| Software, algorithm | FlowJo v9 | FlowJo.com | RRID:SCR_008520 | |
| Software, algorithm | JVenn | jvenn.toulouse.inrae.fr | RRID:SCR_016343 | |
| Software, algorithm | Webgestalt | https://www.webgestalt.org/ | RRID:SCR_006786 | |
| Other | Optimem | ThermoFisher Scientific | 15392402 | Serum free medium |
| Other | Escort IV transfection reagent | Merck | L3287 | Transfection reagent |
| Other | Endothelial cell growth medium 2 | PromoCell | C-22011 | Culture medium |
| Other | Osteosoft | Merck | 1.01728 | Decalcifying reagent for histopathology |
| Other | Heparinase III (EC4.2.2.8 from *Flavobacterium heparinum*) | Merck | H8891 | Enzyme used at 1mU/ml |
| Other | Chondroitinase ABC (EC 4.2.2.4 from *Proteus vulgaris*) | AMSBIO | 100330–1 A | Enzyme used at 10mU/ml |
| Other | Immobilon western chemiluminescence HRP substrate | ThermoFisher Scientific | 11556345 | |
| Other | TRITC-conjugated phalloidin | Merck | FAK100 | Stain for F-Actin used at 1:500 |
| Other | FITC-dextran 70 kDa | Merck | 46945 | Fluorescently-labelled dextran used for permeability studies used at 1:500 |

## Mycolactone and other translocation inhibitors

For all experiments in main figure panels, we used synthetic mycolactone A/B (*Song et al., 2002*), which was generously donated by Prof. Yoshito Kishi (Harvard University). However, we also compared synthetic mycolactone A/B to that extracted from cultures of *M. ulcerans* (which also makes mycolactone A/B) which was a kind gift from Dr Estelle Marion (INSERM). Here, *M. ulcerans* 1615 strain, was grown in Middlebrook 7H10 agar supplemented with Oleic Albumin Dextrose Catalase growth supplement. Bacteria were re-suspended in chloroform-methanol (2:1, v/v) and cell debris were removed after centrifugation. Folch extraction was realized by adding 0.2 volume water. The organic phase was dried and phospholipids were precipitated with ice-cold acetone. The acetone-soluble lipids were loaded on a thin layer chromatography plate and eluted with chloroform-methanol-water (90:10:1) solvent as mobile phase. The yellow band with a retention factor of 0.23 was scraped, filtered, evaporated and then resuspended in absolute ethanol. Mycolactone was stored in absolute ethanol at –20 °C in the dark. The amount of purified mycolactone was determined by high-performance liquid chromatography (HPLC) on a C18 column.

Ipomoeassin F and ZIF-80 (Compound 2 in ref 30) were synthesised by Dr Wei Shi. All synthetic compounds were diluted from stock solutions in DMSO (biological mycolactone stock solution was in ethanol) and were used at the minimal inhibitory concentration, which was 10 ng/ml (~13 nM) mycolactone (*Hall et al., 2022*), 400 nM Ipomoessin F (*Zong et al., 2015*) and 20 nM ZIF-80 (*Zong et al., 2020*). To control for potential impact of the DMSO solvent on cell function, DMSO diluted equivalently was used; typically this was 0.02%.

## Cell culture and treatment

Juvenile, single donor human microvascular endothelial cells (HDMEC) and human umbilical vein endothelial cells (HUVEC) (Promocell) were cultured in hVEGF containing Endothelial cell growth medium 2 (Promocell) at 37 °C and 5% $CO_2$. Cells were routinely seeded at a concentration of $1 \times 10^4$ /cm$^2$ in 25 cm$^2$ or 75 cm$^2$ flasks for no more than 15 population doublings. Where used, laminin-511,–411 or –111 (BioLamina, Sweden) were coated on the surface of uncoated 96-well tissue culture plates at 5 µg/mL in PBS at 4 °C overnight, then washed with medium prior to further experiment. Viability assays used either resazurin dye (Sigma Aldrich) or CellEvent (Invitrogen) as described (*Hall et al., 2022*; *Ogbechi et al., 2018*).

## Uniaxial shear stress induction

In order to mimic the conditions of flow, we used an established technique using an orbital shaker (*Warboys et al., 2019*). HDMEC were grown to confluency in 6-well plates then placed on an orbital shaker rotating at 150 rpm for 24 hr. Mycolactone or DMSO were added in triplicate and the cells cultured for an additional 48 hr maximum. All phase contrast and fluorescent images were taken towards the edge of the wells, where cells experience uniaxial shear stress (*Warboys et al., 2019*) using an Etaluma Lumascope 620.

## Time-lapse imaging of live cells

For time-lapse monitoring, endothelial cells were imaged every 30 min using a zenCELL Owl incubator microscope (innoME GmbH, supplied by LabLogic UK) for 48 hr. Time-lapse videos were generated with zencell-owl software (version 3.3, innoME GmbH), and analysed using their proprietary built-in algorithms of relative cell coverage, proportion of detached cells, and total cell numbers. In some cases, images of cells from certain time points were further analysed in Image J (v1.52n) to cell count of rounded cells per field, and/or the proportion and length/width ratio of elongated cells.

## Scratch assay

Endothelial cells were grown to confluency in 24 well plates then single lines were scratched into the monolayer using a p20 pipette tip. Healing of HDMECs was monitored by imaging at various time points up to 24 hr. Each assay was carried out in triplicate wells. Wounded HUVECs were monitored every 15 min by zenCELL Owl microscope (innoME GmbH) for up to 30 hr.

## *Mycobacterium ulcerans* mouse footpad infection model

*Mycobacterium ulcerans* strain Mu_1082 was cultivated on Middlebrook 7H11 agar (Merck) supplemented with 0.2% glycerol (Thermo Fisher Scientific) and 10% OADC (Thermo Fisher Scientific). Several days before inoculation, bacteria were scraped from the plate and resuspended in 10 ml 7H9 medium (Becton Dickinson) containing 0.5% glycerol, 10% OADC and 0.2% Tween-80 (Merck) and incubated shaking with 3 μm glass beads for 3 days at 31 °C. To prepare the inoculum, cultures were allowed to stand for 10 min (to allow clumps to settle) then 1 ml culture was centrifuged at 13,000 x *g* for 2 min. The supernatant was removed, and the pellet resuspended in Dulbecco's PBS (Thermo Fisher Scientific). After measuring the OD600, $3.33 \times 10^7$ bacteria were pelleted and resuspended in 10 ml PBS, to give an inoculum of $\sim 10^5$ cfu/footpad in a volume of 30 μl.

All in vivo procedures were approved by the University of Surrey's AWERB and UK Home office and met relevant animal welfare and biosafety regulatory standards (under PPL PP0344017). ARRIVE guidelines were followed, and the ARRIVE checklist is provided. In this publication we present new histological analysis of archived material from 8 to 9-week-old wild type C57BL/6J female mice (Charles River, UK), which had been inoculated intradermally with 30 μl of the bacterial suspension or vehicle control (PBS) to the left hind footpad, under gas anaesthesia. Mice were maintained under specific pathogen-free conditions at a temperature of 20–24 °C and humidity of 45 to 65% in individually HEPA filtered cages. The mice were acclimatised for >1 week before being transferred into the CL3, where they acclimatised for at least a further 48 hours before infection. They had free access to water and a standard balanced diet, standard bedding and enrichments including a tunnel and nesting material. Infected mice were housed separately from uninfected mice and blinding was not possible as the infection is clearly visible. Infection grade was assessed daily according to the method of Converse (*Converse et al., 2011*), where Grade 1 showed swelling of the metatarsal area (<50% increase compared to normal), Grade 2 showed greater swelling (50-150%) and Grade 3 had swelling further up the leg, visualised at the hock. No mice were excluded, experienced adverse events, or reached the humane endpoint (based on infection score, mobility, weight, and assessment of pain and secondary infection; or oedema extending into the "thigh" area and/or bedding adhering to the footpad suggesting footpad ulceration) in these experiments. Mice were killed by a schedule 1 method (cervical dislocation). The whole foot was then removed and fixed by immersion in 10% neutral buffered formalin for at least 24 hr.

## Histological analysis of murine foot samples

Fixed murine feet were decalcified using the EDTA-based Osteosoft solution (Merck) and then embedded in paraffin for histological analysis by Ziehl-Neelsen stain, Alcian blue-periodic acid Schiff stain, and immunohistochemistry (IHC) for fibrin(ogen). For IHC staining, 5 µm tissue sections on polylysine-coated slides were deparaffinised, endogenous peroxidase quenched, epitope unmasked with heated IHC citrate buffer (pH 6.0) (Merck) and blocked with 5% bovine serum albumin. The tissue sections were incubated with anti-fibrinogen antibody (A0080, DAKO) or matched isotype control overnight at 4 °C. Staining was then performed with biotinylated horse anti-rabbit IgG (Vector Laboratories) and VECTASTAIN Elite ABC kit and ImmPACT NovaRED peroxidase substrate and further counterstained with Harris Haematoxylin (ThermoFisher Scientific). Whole slide images were captured using the NanoZoomer slide scanner (Hamamatsu Photonics) and analysed using ImageScope software (Leica Biosystems) and ndp2.view software (Hamamatsu). Some photographs were taken with Micropix microscope camera (acquisition software Cytocam) attached to a Yenway CX40 laboratory microscope (Micropix).

## Electron microscopy

Glabrous skin from the infected hind paw of a mouse at grade 1 was dissected and freshly immersed in a fixative solution containing 4% Formaldehyde and 3% Glutaraldehyde. After 24 hr, the tissue area of interest was chosen via examination of semi-thin sections of 500 nm stained with toluidine blue. By using a Leica ultramicrotome with diamond knife, ultrathin sections of 100 nm were collected into copper grids and images were obtained by FEI Tecnai T12.

## Membrane protein preparation

HDMEC ($1 \times 10^7$ cells) were seeded onto 15 cm dishes (Corning) and grown to 90% confluency then exposed to solvent carrier DMSO or 10 ng/ml mycolactone for 24 hr. Cells were washed four times in PBS and once in lysis buffer (10 mM Tris pH 7.5, 250 mM Sucrose, protease inhibitor cocktail). Cells were incubated for 20 min on ice in 10 ml lysis buffer then lysed in by 20 strokes dounce homogenisation. Lysates were centrifuged at 1000xg for 10 min at 4 °C then the post-nuclear supernatant was centrifuged at 100,000xg for 1 hr at 4 °C. Pellets were resuspended in 110 µl lysis buffer. Protein concentration was determined by BCA assay and 50 µg aliquots were subjected to acetone precipitation. Triplicate samples were prepared from three independent assays.

## Proteomics

Acetone precipitated proteins were reduced, alkylated and digested with trypsin before 9-plex isobaric TMT labelling according to the manufacturer's protocol. Labelled peptides were separated by high pH reverse phase liquid chromatography, collecting 20 fractions which were then lyophilised, desalted and analysed by LC-MS/MS. TMT labelled samples were analysed by the SPS-MP3 method using an Orbitrap Lumos mass spectrometer. Spectra were searched using the Mascot search engine version (Matrix Science) and analysed using the Proteome Discovery platform. (Version 2.2 Thermo Fisher Scientific). NA values and low confidence proteins were removed, and data was normalised using each channel median. Differential expression analysis was carried out using Limma. Adjusted p values were calculated by the Benjamini-Hochberg method. UniProt and the Human Protein Atlas (https://www.proteinatlas.org) were used to determine protein location and characteristics. Over-representation of GO groups was assessed using Webgestalt (https://www.webgestalt.org/) (ref). Signal peptide ΔG values were obtained via the ΔG Prediction Server V1.0 (https://dgpred.cbr.su.se). The mass spectrometry proteomics data have been deposited to the ProteomeXchange Consortium via the PRIDE (*Perez-Riverol et al., 2022*) partner repository with the dataset identifier PXD037489.

## siRNA transfection

HUVECs ($2 \times 10^5$ cells) were seeded onto a six-well plate. The next day, cells were washed with Opti-MEM (Gibco) and then kept in 1 ml medium. Each siRNA (Silencer siRNA assays ID#112321, #112322 for B3GALT6 or Silencer negative control No.1 siRNA AM4611; Invitrogen ThermoFisher Scientific) was diluted in Opti-MEM to 0.6 µM, mixed with equal volume of diluted Escort IV transfection reagent (L3287, Merck; final concentration 60 µg/mL). The transfection was performed onto HUVECs with ~40% confluency. Medium was changed back to normal endothelial culture medium 5 hr later. The

transfectants received treatment 24 hr post transfection and were subjected to the respective analysis (e.g. image, sample harvest for immunochemical assays) after another 24 hr.

## Flow cytometry

Flow cytometry was carried out according to standard methods as described in *Ogbechi et al., 2015* using an Attune NxT flow cytometer (ThermoFisher Scientific). Cells were detached with non-enzymatic cell dissociation solution (Merck) or briefly (for Itgb4 staining only) trypsinised with 0.04% trypsin/ 0.03% EDTA (PromoCell). For surface GAG detection, cells were treated with 1mU of heparinase III (EC4.2.2.8 from *Flavobacterium heparinum*) to expose the neo-epitope of heparan sulphate or with chondroitinase ABC (EC 4.2.2.4 from *Proteus vulgaris*) (AMS Biotechnology) for 1 hr at 37 °C prior to the staining procedures. Antibodies were Δ-HS (F69-3G10, AMS Biotechnology), CS (CS56, Merck), HSPG2/perlecan (7B5, ThermoFisher Scientific), glypican-1 (AF4519), integrin β4/CD104 (clone 439-9B, eBioscience), integrin β1/CD29 (P4C10, NBP2-36561), syndecan-2 (MAB2965), biglycan (AF2667), laminin α5 (NBP2-42391) from Biotechne. Isotype control mouse IgG1 (P3.6.2.8.1; 14-4714-81 from Invitrogen), mouse IgG2b (MG2B00), goat IgG (AB-108-C from R&D), rat IgG2b (14-4031-81), mouse IgM (PFR-03) and fluorophore-conjugated secondary antibodies goat anti-mouse IgG PE (12-4010-82), donkey anti-goat IgG FITC (A16000), and anti-rat IgG FITC (31629) were from ThermoFisher Scientific. The main population was gated by forward and side scatter plot of untreated cells using FlowJo (v9); among this, single cell population of $10^4$ cells per condition was subjected to analysis. Mean fluorescence intensity was determined and presented as % relative to untreated control.

## Immunoblotting

Immunoblotting was carried out according to standard methods as described in *Ogbechi et al., 2015*. Endothelial cells were lysed either in RIPA buffer (where protein content-equalised post-nuclear fractions were used) or directly in 'gel sample buffer' (with sonication to degrade genomic DNA). Immunoblotting of commercial pre-cast gels (BioRad) used either Immobilon PVDF membranes (Merck) or nitrocellulose membranes (GE Healthcare). Antibodies used in this study were: Δ-HS (F69-3G10, AMS Biotechnology); anti-fibronectin (AB1945, Merck); anti-integrin α5 (sc-166665); anti-rabbit-HRP (GE Healthcare, NA934V), anti-mouse-HRP (GE Healthcare, NA931V). To visualise the HS neoepitope, protein lysate was digested with 1 mU of heparinase III (EC4.2.2.8 from *Flavobacterium heparinum*) prior to SDS-PAGE. Blots were developed using enhanced chemiluminescence with Immobilon western chemiluminescence HRP substrate (Thermo Fisher Scientific, 11556345) and imaged on a Fusion FX Imager (Vilber-Lourmatclean), which provides a warning if the areas of the image are saturated.

## Immunofluorescence

Immunofluorescent imaging was carried out according to standard methods as described in *Hall et al., 2022*. Cells were fixed with 4% paraformaldehyde in PBS. For visualising intracellular markers, cells were permeabilised wit 0.25% Nonidet P-40 alternative in NETGEL buffer (150 mM NaCl, 5 mM EDTA, 50 mM Tris-Cl, pH 7.4, 0.05% Nonidet P-40 alternative, 0.25% gelatin and 0.02% sodium azide). Antibodies used in this study were: B3GALT6 (H00126792-B01P, Biotechne), GOLGB1/Giantin (ab80864, abcam), Laminin α4 (AF7340, Biotechne), Δ-HS (F69-3G10, AMS Biotechnology), HSPG2/perlecan (7B5, Thermo Fisher Scientific), TRITC-conjugated phalloidin (FAK100, Merck), Alexa Fluor 594 goat anti-rabbit (A11012), Alexa Fluor 488 donkey anti-mouse (A21202) and Alexa Fluor 488 donkey anti-sheep (A11015) from Invitrogen/Thermo Fisher Scientific. For B3GALT6 intensity in the Golgi apparatus, the region of interest per cell was defined by giantin-positive staining using ImageJ selection tools. The integrated density of B3GALT6 fluorescence of selected regions and background reading were then measured and the difference between the two numbers were corrected total cell fluorescence.

## Vascular permeability assay

Endothelial cells were seeded on hanging cell culture inserts containing 1 μm pores with a polyethylene terephthalate membrane (Falcon). Treatment as above or with 100 ng/mL IL-1β (Gibco) were applied to both the insert and receiver wells. After indicated time, fluorescein isothiocyanate (FITC)-conjugated dextran (70 kDa, Millipore) was applied to each insert for 20 min. The fluorescence intensity of the solution in the receiver wells was then assessed by a fluorescent plate reader (FLUOstar

Omega, BMG Labtech) with excitation/ emission wavelength at 485/530 nm. Fluorescence intensity was normalised to untreated control wells with an intact monolayer of endothelial cells and expressed as a % of subtracted value obtained from wells where no cells were seeded to the insert.

### Adhesion assay

HDMECs were harvested, incubated with anti-integrin β1 (clone P4C10, NBP2-36561, Biotechne) or isotype control mouse IgG1 (P3.6.2.8.1; 14-4714-81 from Invitrogen) for 5 min, then $1.5 \times 10^4$ cells were added to the wells of a 96-well plate that had been coated or not with different laminins as described above. After 1 hr, each well was washed three times with serum free medium and attached cells were imaged with a digital microscope camera (Micropix) attached to an AE31E inverted microscope (Motic). The cell count per image was determined using ImageJ.

### Statistical analysis

All data, with exception of the live-cell imaging using the ZenCell OWL (see above), were analysed using GraphPad Prism Version 9.4.1 and 10.2.3. Data were analysed using a one- or two-way ANOVA using an appropriate correction for multiple comparisons (either Dunnett's, or Tukey's). Some two-way ANOVAs also included the Geisser Greenhouse correction for sphericity.

## Acknowledgements

We are extremely grateful to Prof Yoshito Kishi (Harvard University, USA) for the gift of synthetic mycolactone A/B. We dedicate this paper to the memory of this tremendous scientist who did some much to advance research into the pathogenesis of Buruli ulcer. We thank Dr Estelle Marion (Inserm U1302 INCIT, Angers, France) for the biologically purified mycolactone, and Prof Richard Phillips (Kumasi Centre for Collaborative Research in Tropical Medicine) who provided the M ulcerans strain used in this work. We thank Katherine Walker and Ella May of the University of Surrey's Veterinary Pathology Centre for their assistance with tissue processing and staining of murine samples, and Dr Paola Campagnola (University of Surrey) for her advice in setting up the endothelial cells under flow. This work is supported by a Wellcome Trust Investigator Award in Science to Prof Rachel Simmonds (WT202843/Z/16/Z). Chemical synthesis of Ipomoeassin F and ZIF-80 was supported by an AREA grant GM116032 from the National Institute of General Medical Sciences of the National Institutes of Health (NIH) to Prof Wei Q Shi. Belinda Hall was supported by Medical Research Council grant MR/W02618X/1 to Prof Rachel Simmonds.

## Additional information

### Funding

| Funder | Grant reference number | Author |
|---|---|---|
| Wellcome Trust | WT202843/Z/16/Z | Rachel E Simmonds |
| National Institute of General Medical Sciences | GM116032 | Wei Q Shi |
| Medical Research Council | MR/W02618X/1 | Rachel E Simmonds Belinda S Hall |

The funders had no role in study design, data collection and interpretation, or the decision to submit the work for publication. For the purpose of Open Access, the authors have applied a CC BY public copyright license to any Author Accepted Manuscript version arising from this submission.

### Author contributions

Louise Tzung-Harn Hsieh, Data curation, Formal analysis, Validation, Investigation, Visualization, Methodology, Writing – original draft; Belinda S Hall, Data curation, Formal analysis, Investigation, Visualization, Writing – original draft, Writing – review and editing; Jane Newcombe, Tom A Mendum, Investigation, Methodology; Sonia Santana Varela, Formal analysis, Investigation, Visualization, Writing

– review and editing; Yagnesh Umrania, Formal analysis, Investigation, Methodology; Michael J Deery, Formal analysis, Supervision, Investigation, Methodology; Wei Q Shi, Resources; Josué Diaz-Delgado, Formal analysis, Writing – review and editing; Francisco J Salguero, Data curation, Formal analysis, Writing – original draft; Rachel E Simmonds, Conceptualization, Resources, Data curation, Formal analysis, Supervision, Funding acquisition, Visualization, Project administration, Writing – review and editing

## Author ORCIDs

Louise Tzung-Harn Hsieh (iD) https://orcid.org/0000-0002-9499-889X
Belinda S Hall (iD) http://orcid.org/0000-0002-3753-1978
Rachel E Simmonds (iD) https://orcid.org/0000-0003-4843-8266

## Ethics

All in vivo procedures were approved by the University of Surrey's AWERB and UK Home office and met relevant animal welfare and biosafety regulatory standards (under PPL PP0344017). Animals were housed in cages with environmental enrichments in a unit with a strong culture of care. Animals were infected under gas anaesthesia. ARRIVE guidelines were followed, and the ARRIVE checklist is provided.

Reviewer #1 (Public review): https://doi.org/10.7554/eLife.86931.3.sa1
Reviewer #2 (Public review): https://doi.org/10.7554/eLife.86931.3.sa2
Author response https://doi.org/10.7554/eLife.86931.3.sa3

# Additional files

## Supplementary files

MDAR checklist

## Data availability

The mass spectrometry proteomics data have been deposited to the ProteomeXchange Consortium via the PRIDE partner repository with the dataset identifier PXD037489. Figure 1—source data 1, Figure 4—source data 1, Figure 5—source data 1, Figure 6—source data 1, Figure 7—source data 1, and Figure 8—source data 1 contain the numerical data used to generate graphs.

The following dataset was generated:

| Author(s) | Year | Dataset title | Dataset URL | Database and Identifier |
|---|---|---|---|---|
| Hall BS, Umrania Y, Deery MJ, Simmonds RE | 2025 | Mycolactone causes destructive Sec61-dependent loss of the endothelial glycocalyx and basement membrane: a new indirect mechanism driving tissue necrosis in Mycobacterium ulcerans infection | http://www.ebi.ac.uk/pride/archive/projects/PXD037489 | PRIDE, PXD037489 |

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
