## [Editor Report · eLife Assessment]

The toxin mycolactone is produced by Mycobacterium ulcerans which is responsible for the Buruli ulcer lesions. The authors performed a **valuable** study showing the effects of mycolactone on blood vessel integrity. This **convincing** data provides new therapeutic targets to accelerate the healing of Buruli ulcer lesions.

---

## [Referee Report · Reviewer #1 (Public review)]

Summary:

By employing human primary microvascular endothelial cells, along with live confocal imaging, proteomics, and chemical validation studies, the authors reveal a novel cellular mechanism underlying mycolactone's effects in Buruli ulcer lesions. This finding provides important insights into the specific mechanisms of skin pathogenesis.

Strengths:

The techniques employed are state-of-the-art.

Weaknesses:

The study lacks genetic validation of the findings.

---

## [Referee Report · Reviewer #2 (Public review)]

The authors have investigated the effect of the toxin mycolactone produced by Mycobacterium ulcerans on the endothelium. Mycobacterium ulcerans is involved in Buruli ulcer lesions classified as a neglected disease by WHO. This disease has dramatic consequences on the microcirculation causing important cutaneous lesions. The authors have previously demonstrated that endothelial cells are especially sensitive to mycolactone. The present study brings more insight into the mechanism involved in mycolactone-induced endothelial cells defect and thus in microcirculatory dysfunction. The authors showed that mycolactone directly affected the synthesis of proteoglycans at the level of the golgi with a major consequence on the quality of the glycocalyx and thus on the endothelial function and structure. Importantly, the authors show that blockade of the enzyme involve in this synthesis (galactosyltransferase II) phenocopied the effects of mycolactone. The effect of mycolactone on the endothelium was confirmed in vivo. Finally, the authors showed that exogenous laminin-511 reversed the effects of mycolactone, thus opening an important therapeutic perspective for the treatment of wound healing in patients suffering Buruli ulcer lesions.

---

## [Author Response]

The following is the authors’ response to the current reviews.

**Response to reviewer 1:**

We thank the reviewer for their positive comments and note that we made many attempts to genetically alter endothelial cells to expression mutants of SEC61A1 that are resistant to the effects of mycolactone. However, these cells were not capable of supporting expression of this transgene. Instead, we used an approach where we tested other translocation inhibitors, with a different chemical structure but same mechanism of action at the Sec61 translocon and found that these phenocopied the effects.

The following is the authors’ response to the original reviews.

**Public Reviews:**

**Reviewer #1 (Public Review):**
The authors have investigated the effect of the toxin mycolactone produced by mycobacterium ulcerans on the endothelium. Mycobacterium ulcerans is involved in Buruli ulcer classified as a neglected disease by WHO. This disease has dramatic consequences on the microcirculation causing important cutaneous lesions. The authors have previously demonstrated that endothelial cells are especially sensitive to mycolactone. The present study brings more insight into the mechanism involved in mycolactone-induced endothelial cells defect and thus in microcirculatory dysfunction. The authors showed that mycolactone directly affected the synthesis of proteoglycans at the level of the golgi with a major consequence on the quality of the glycocalyx and thus on the endothelial function and structure. Importantly, the authors show that blockade of the enzyme involve in this synthesis (galactosyltransferase II) phenocopied the effects of mycolactone. The effect of mycolactone on the endothelium was confirmed in vivo. Finally, the authors showed that exogenous laminin-511 reversed the effects of mycolactone, thus opening an important therapeutic perspective for the treatment of wound healing in patients suffering Buruli ulcer and presenting lesions.
**Reviewer #2 (Public Review):**
The authors dissected the effects of mycolacton on endothelial cell biology and vessel integrity. The study follows up on previous work by the same group, which highlighted alterations in vascular permeability and coagulation in patients with Buruli ulcer. It provides a mechanistic explanation for these clinical observations, and suggests that blockade of Sec61 in endothelial cells contributes to tissue necrosis and slow wound healing.Overall, the generated data support their conclusions and I only have two major criticisms:- Replicating the effects of mycolactone on endothelial parameters with Ipomoeassin F (or its derivative ZIF-80) does not demonstrate that these effects are due to Sec61 blockade. This would require genetic proof, using for example endothelial cells expressing Sec61A mutants that confer resistance to mycolactone blockade. The authors claimed in the Discussion that they could not express such mutants in primary endothelial cells, but did they try expressing mutants in HUVEC cell lines? Without such genetic evidence all statements claiming a causative link between the observed effects on endothelial parameters and Sec61 blockade should be removed or rephrased. The same applies to speculations on the role of Sec61 in epithelial migration defects in discussion. Data corresponding to Ipomoeassin F and ZIF-80 do not add important information, and may be removed or shown as supplemental information.- While statistical analysis is done and P values are provided, no information is given on the statistical tests used, neither in methods nor results. This must be corrected, to evaluate the repeatability and reproducibility of their data.

We respectfully but fundamentally disagree with the comments regarding the Sec61 dependence of the effects that we observed. We showed that loss of glycocalyx and basement membrane components underpinned the phenotypic changes in endothelial cells (morphological changes, loss of adhesion, increased permeability, and reduced ability to repair scratch wounds). We demonstrated that we could phenocopy permeability increases and elongation phenotype by knocking down the type II membrane protein B3Galt6, and reverse the adhesion defect by exogenous provision of the secreted laminin-511 heterotrimer.

Our conclusion that mycolactone mediates these effects via Sec61 inhibition is not based solely on the use of alternative inhibitors but is built on several pillars of evidence:

First, the proteomics data conforms entirely to predictions based on the topology of affected vs. non-effected proteins, and agrees with independently published proteomic datasets from T lymphocytes, dendritic cells and sensory neurons (ref.12), as well as biochemical studies performed using in vitro translocation assays (ref.11,34). Furthermore, the pattern of membrane protein down regulation observed in our experiments fits perfectly with established models of protein translocation mechanisms, particularly with respect to the lack of effect on specific topologies of multipass membrane proteins, tail anchored- and type III membrane proteins (ref.34-36).

Second, since Sec61 very highly conserved amongst mammals and is found in all nucleated cells, it is hard to conceptualise a framework in which mycolactone targets Sec61 in some cells and not others, as this reviewer suggests might be the case for epithelial cells [noting that the work being referred to (ref.29) predates our 2014 work showing that mycolactone is a Sec61 inhibitor (ref.7)]. Indeed, mycolactone has been shown to target Sec61 in multiple independent approaches including forward genetic screens involving random mutagenesis and CRISPR/Cas9 (ref.10, PMID: 35939511). Genetic evidence has previously been provided for the Sec61 dependence of mycolactone effects in epithelial cells (ref.10,17). We have unpublished genetic evidence that the rounding and detachment of epithelial cells due to mycolactone is reduced when resistance mutations are over expressed, and will consider including this in the next version of the manuscript.

Third, given this weight of evidence, one would be hard-pressed to provide an alternative explanation for the specific down-regulation of glycosaminoglycan-synthesising enzymes and adhesion/basement membrane molecules while most cytosolic and non-Sec61 dependent membrane proteins are unchanged or upregulated. However, seeking to be as rigorous as possible we have here shown that a completely independent Sec61 inhibitor produces the same phenotype at the gross and molecular level. Ipomoeassin F (Ipom-F) is a glycolipid, not a polyketide lactone, yet they both compete for binding with cotransin in Sec61α (ref.6). There is significant overlap in the cellular responses to mycolactone and Ipom-F, including the induction of the integrated stress response (ref.17, PMID: 34079010), which we observed again in the current data, providing further evidence that this approach is useful when genetic approaches are technically unattainable.

Therefore, we are confident the effects seen on endothelial cells are Sec61-dependent. We are happy to provide more detail on our lengthy attempts at over-expressing mycolactone resistant SEC61A1 genes in HUVECs; primary endothelial cells derived from the umbilical vein. We are highly experienced in this area, and have previously stably expressed these proteins in epithelial cell lines, reproducing the resistance profile (ref.10,17). Notably though, these cells do not have normal ‘fitness’ in the absence of challenge. Since endothelial cells (and endothelial cell lines; PMID: 12560236) are extremely hard to transfect with plasmids, with efficiency routinely 5-10% (including in our hands), we developed a lentivirus system. We were eventually (after multiple attempts using different protocols) able to transduce primary HUVECs with constructs expressing GFP (at an efficiency of about 10-20%) and select/expand these under puromycin selection. Never-the-less, we never recovered any cells that expressed the flag-tagged SEC61A1 wild type or SEC61A1 carrying the resistance mutant D60G. We also attempted to select D60G-transduced cells with mycolactone epimers, an approach that can help the cells compete against non-transduced cells in culture flasks (ref.10). We concluded that primary endothelial cells are unable to tolerate the expression of additional Sec61α, and this was incompatible with survival.

It’s also important to note that most endothelial cell specialists would agree that endothelial cell lines are not good models of endothelial behaviour. We tested the HMEC-1 cell line, but found it did not express prototypical endothelial marker vWF in the expected way. Therefore we focussed our efforts on primary endothelial cells. Should we be able to overcome the dual challenge of the necessity to work in primary cells, and the difficulty of over-expressing Sec61, we will update this paper at a later date with this data, and will also expand the above arguments.

We apologise for the embarrassing oversight of not including information about the statistical analyses we used, which of course we will correct in full in the revised version. However, we would like to provide this information to readers of the current version of the manuscript. All data were analysed using GraphPad Prism Version 9.4.1:

Figure 1: one-way ANOVA with Dunnett’s (panel A) or Tukey’s (panel B) correction for multiple comparisons

Figure 2 supplement: one-way ANOVA with Tukey’s correction for multiple comparisons (analysed panel)

Figure 3: one-way ANOVA with Tukey’s (panel B) or Dunnett’s (panel E&F) correction for multiple comparisons

Figure 4: one-way ANOVA with Dunnett’s correction for multiple comparisons (all analysed panels)

Figure 5 and supplement: one-way ANOVA with Dunnett’s correction for multiple comparisons (all analysed panels)

Figure 6: one-way ANOVA with Dunnett’s correction for multiple comparisons (analysed panel)

Figure 6 supplement: one-way ANOVA with Dunnett’s correction for multiple comparisons (all analysed panels)

Figure 7: two-way ANOVA with Tukey’s correction for multiple comparisons (all analysed panels; panels B&C also included the Geisser Greenhouse correction for sphericity)

Figure 7 supplement: Panels A&D used a repeated measures one-way ANOVA with Dunnett’s correction for multiple comparisons (panel D also included the Geisser Greenhouse correction for sphericity). Panels B,C&E used a two-way ANOVA with Tukey’s correction for multiple comparisons (panels B&C also included the Geisser Greenhouse correction for sphericity)

**Reviewer #3 (Public Review):**
Buruli ulcer is a severe skin infection in humans that is caused by a bacterium, Mycobacterium ulcerans. The main clinical sign is a massive tissue necrosis subsequent to an edema stage. The main virulence factor called mycolactone is a polyketide with a lactone core and a long alkyl chain that is released within vesicles by the bacterium. Mycolactone was already shown to account for several disease phenotypes characteristic of Buruli ulcer, for instance tissue necrosis, host immune response modulation and local analgesia. A large number of cellular pathways in various cell types was reported to be impacted by mycolactone. Among those, the Sec61 translocon involved in the transport of certain proteins to the endoplasmic reticulum was first identified by the authors of the study and is currently the most consensual target. Mycolactone disruption of Sec61 function was then shown to directly impact on cell apoptosis in macrophages, limited immune responses by T-cells and increased autophagy in dermal endothelial cells and fibroblasts. In their manuscript, TzungHarn Hsieh and their collaborators investigated the Sec61- dependent role of mycolactone on morphology, adhesion and migration of primary human dermal microvascular endothelial cells (HDMEC). They used a combination of sugar and proteomic studies on a live imagebased phenotypic assay on HDMEC to characterize the effect of mycolactone. First, they showed that upon incubation of monolayer of HDMEC with mycolactone at low dose (10 ng/mL) for 24h, the cells become elongated before rounding and eventually detached from the culture dish at 48h. Next, mycolactone was probed on a scratch assay and migration of the cells ceased upon a 24h incubation. The same effect as mycolactone on these two assays was observed for two other Sec61 inhibitors Ipomoeassin F and ZIF-80. Then, the authors resorted to the widely established mouse footpad model of M. ulcerans infection to evidence fibrinogen accumulation outside the blood vessel within the endothelium at 28 days postinfection, correlating with severe endothelial cell morphology changes.To dissect the molecular pathways involved in these phenotypes, the authors performed an HDMEC membrane protein analysis and showed a decrease in the numbers of proteins involved in glycosylation and adhesion. As protein glycosylation mainly occurs in the Golgi apparatus, a deeper analysis revealed that enzymes involved in glycosaminoglycan (GAG) synthesis were lost in mycolactone treated HDMEC. A combination of immunofluorescence and flow cytometry approaches confirmed the impact of mycolactone on the ability of endothelial cells to synthesize GAG chains. The mycolactone effect on cell elongation was phenocopied by knock-down of galactosyltransferase II (B3Galt6) involved in GAG biosynthesis. A second extensive analysis of the endothelial basement membrane component and their ligands identified multiple laminins affected by mycolactone. Using similar functional studies as for GAG, the impact of mycolactone on cell rounding and migration could be reversed by the addition of laminin α5.The major strengths of the study relies on a combination of cleverly designed phenotypic assays and in-depth cleverly designed membrane proteomic studies and follow-up analysis.The results really support the conclusions. Congratulations!The discussion takes into account the current state of the art, which has mostly been established by the authors of the present manuscript.
**Recommendations for the authors:**

In preparing this revised version we have made a number of general improvements:

• We added the missing information on statistical analysis that was mentioned in the public review of reviewer #2

• We have changed all gene names to the HUGO nomenclature

• We have changed our abbreviation of mycolactone from “MYC” to “Myco” in all figures to avoid any potential confusion with other protein factors

• We have moved the fibrin(ogen) staining of the mouse footpads to its own figure (now Fig 2), partly due to the inclusion of additional data in Fig 1. This has changed the numbering of subsequent figures, but has also made the supplementary figures easier to track.

**Reviewer #1 (Recommendations For The Authors):**
(1) Figure 1I. When mice are injected M. Ulcerens a measurement of local blood flow would be very informative in addition of the data shown. Cutaneous blood flow at the level of the feet is possible using laser doppler or Laser speckle imaging. With these measurements the authors would have a functional quantification of the effect of the glycosaminoglycans- Sec61α associated damages on the microcirculatory blood flow. The same measurement could also better validate the therapeutic effect of laminin.

We thank the reviewer for this great suggestion, and respectfully remind the reviewer that these experiments take place in CL3 containment. This often completely precludes certain procedures due to the availability of equipment inside the containment, and our ability to sterilise it. Where we are able to perform procedures, it greatly increases their complexity since any procedures on live animals must take place inside of a cabinet. Therefore, we can only use equipment that we have at our animal facility. It is not trivial to set up the regulatory permissions to perform these experiments at other facilities where more specialist equipment is located due to the containment restrictions.

Never-the-less we have attempted to perform ultrasound imaging of mouse feet using the VivoF and have set up a collaboration with other researchers at Surrey who have developed a novel imaging instrument to measure microvascular circulation call optical coherence tomography (OCT; https://pubmed.ncbi.nlm.nih.gov/34882760/), and we are working with them to develop a protocol that be used in small rodents.

However, while we have dedicated considerable time to trying to perform the suggested experiment, we have not been successful within a reasonable time frame. Consequently, if we are able to establish this technique in the *M. ulcerans* infection model, and/or OCT in small rodents, this will likely be beyond the scope of the current manuscript and will be a publication in its own right. We note that we have been able to perform almost all of the other requested experiments (see below), and have also been able to undertake transmission electron microscopy of *M. ulcerans* infected mouse footpads, which confirms the loss of the basement membrane at high resolution (Fig 7E).

(2) Figure 1 -D. Endothelial cells were exposed to mycolactone, Ipomoeassin F or ZIF-80. The effect on the cells is clear and impressive. Nevertheless, endothelial cells in no flow conditions are considered "diseased" cells as in the areas of low flow or no flow are prone to atherosclerosis in vivo. Would the authors expect similar effects in cells submitted to flow? In this conditions cells would be already elongated in the direction of flow.

We agree that flow is usually experienced by endothelial cells *in vivo*, and have repeated a selection of our experiments under conditions that mimic flow and produce uniaxial shear stress. All showed a similar pattern of response to mycolactone, including the phenotypic changes (Fig 1I-K), loss of perlecan (Fig S6C) and laminin α4 (Fig S7B). It is true that the elongation phenotype is not as striking in a cell monolayer that already contains many elongated cells, but qualitatively the cells become disorganised and at 48 hours, their length/width ratio had increase. These results provide reassurance that our findings are physiologically relevant.

(3) Discuss the possible consequences of your findings on vascular reactivity and especially on flow-mediated dilation and/or flow-mediated remodeling which as both are important in tissue repair and wound healing.

We agree with this reviewer that there are likely to be broad consequences to endothelial and vascular function as a result of our findings here. Vascular reactivity is not something we directly considered in this manuscript, and is probably better linked to our planned future work, laid out above, regarding vascular flow in the infected animals. While a key mediator of vascular tone, endothelin 1, is a Sec61-dependent secreted peptide mediator (and is likely to also be affected by mycolactone’s actions), this was not one of the >6500 proteins we identified in our proteomic study. On the other hand, it has been shown by others that mycolactone can induce NO production by in other types of cells.

**Reviewer #2 (Recommendations For The Authors):**
- The authors use a mouse model of M. ulcerans infection of footpads to assess the in vivo relevance of their results. It would be useful to comment on any differences between human and mouse with regard to endothelial cell biology and vessel wall architecture. Since the authors have access to patients samples, parallel stainings in human lesions would have strengthened the study.

This is an important issue, and is one we have already addressed in our two previous articles https://pubmed.ncbi.nlm.nih.gov/35100311/
https://pubmed.ncbi.nlm.nih.gov/26181660/ . Indeed, this latter work already included a detailed analysis of fibrin staining in these Buruli ulcer patient biopsies and underpinned the hypothesis that we have now tested in the current manuscript.

It is worth noting that our data supports that the critical step is at an early (pre-clinical) stage, for which patient samples are not available. The proposed human challenge model (https://pubmed.ncbi.nlm.nih.gov/37384606/) may well provide a suitable platform such studies in the future.

- The authors should provide in the Discussion some explanation for the differential effects of Laminin-11, -411 and -511 in Fig. 7

This is an interesting point, and probably related to the expression of laminin binding proteins by mycolactone-exposed endothelial cells. We pursued several candidates based on the proteomic data but could not identify a unique gene that explained this observation. Mostly likely they are explained by partial (be it low or high) loss of a combination of integrin binding proteins. Since this was rather inconclusive and we preferred not to present this data, and already said (p34-35) “We have not been able to ascribe this to the retention of a specific adhesion molecule, and instead postulate that rescue could be via residual expression of a wide variety of laminin α5 receptors.”

- The word "catastrophic" in the title is very dramatic given the limited impact on the vital prognosis of patients

This word has been changed to “destructive”

**Reviewer #3 (Recommendations For The Authors):**

Several points could be further discussed:

-In mouse model of M. ulcerans infection, in 5% of cases, animals heal spontaneously. How could the authors results contribute to bring hypothesis to this phenomenon?

Others have shown that the ability of some mice to control *M. ulcerans* infection is related to loss of mycolactone production by an unknown mechanism. It is not something we have ever observed in the infection experiments we have performed, although this may be due to the humane endpoints of our licence. However, this seems somewhat outside the main focus of the paper and we have not discussed this further.

-Mycolactone was also reported to induce analgesia in the mouse model. There is still controversy about the precise mechanisms involved in this mycolactone mediated painless effect. Could the data obtained here help to resolve the controversy?

We agree that analgaesia in *M. ulcerans* infection (both in mouse models and in clinical infections) is an extremely interesting area. However, we cannot mechanistically link loss of vascular integrity with the analgaesia based on the data generated in the current manuscript. Therefore we prefer not to speculate on this.

The quantification of the microscopy images and videos should be provided as well as the script used to quantify them.

The reviewer is not specific about which microscopy images are being referred to in this comment, but the reference to videos leads us to assume this is related to the ZenCell OWL images/videos presented in Figure 1 and Figure S1. We had already provided quantification of these in the graphs provided, and the algorithms use for % coverage and % detached cells were provided in the instrument software used to gather the data, the ZenCell OWL (which are proprietary). Other counts were made manually, and the length:width ratio is simple arithmetic as already described in the methodology.

The authors performed their work using chemically synthesized mycolactone obtained from the very generous Professor Kishi (Harvard University). Would the same phenotype and proteomics analysis be obtained with biologically purified mycolactone?

Our lab has extensive experience of both biologically purified and synthetic mycolactone, and the phenotypes observed have always been identical when using the chemically synthesised form. Therefore we did not repeat the proteomics experiments as we do not believe it would provide any greater insight into the disease mechanism. However, we have now replicated a range of findings using mycolactone biologically purified from *M. ulcerans*. In particular, we confirmed that the cytotoxic activity of synthetic and biological mycolactone are inseparable (Figure S1A), and the main phenotypic changes induced by mycolactone in endothelial cells (Phenotypes; Figures S1D-F, B3GALT6/perlecan/laminin α5 loss; S5A, S6B, S7A).

Although already very comprehensive, a kinetic study of their proteomic analysis over time could strengthen the analysis (from 2H to 48H).

We agree that more data is always better, but since we validated our proteomic data set over multiple timepoints between 2 and 48 hrs, we do not believe this would alter the main conclusions of our work.

The siRNA transfection protocol could be better described. A Table listing all the reagents would help the reader.

A more detailed siRNA transfection protocol has been added to the methods section, and we now include a Key Resources Table at the start of the Materials & Methods section.